# The ToMCAT Dataset

**Adarsh Pyarelal**[1], **Eric Duong**[2], **Caleb Jones Shibu**[2], **Paulo Soares**[2], **Savannah Boyd**[3],
**Payal Khosla**[4], **Valeria Pfeifer**[3], **Diheng Zhang**[3], **Eric Andrews**[3], **Rick Champlin**[1],
**Vincent Raymond**[5], **Meghavarshini Krishnaswamy**[6], **Clayton Morrison**[1],
**Emily Butler**[4], **Kobus Barnard**[2]

[1]School of Information, [2]Department of Computer Science, [3]Department of Psychology,
[4]Norton School of Human Ecology, [5]Lum AI, [6]Department of Linguistics
University of Arizona
adarsh@arizona.edu
**https://tomcat.ivilab.org**

## Abstract

We present a rich, multimodal dataset consisting of data from 40 teams of three humans conducting simulated urban search-and-rescue (SAR) missions in a Minecraft-based testbed, collected for the Theory of Mind-based Cognitive Architecture for Teams (ToMCAT) project. Modalities include two kinds of brain scan data—functional near-infrared spectroscopy (fNIRS) and electroencephalography (EEG), as well as skin conductance, heart rate, eye tracking, face images, spoken dialog audio data with automatic speech recognition (ASR) transcriptions, game screenshots, gameplay data, game performance data, demographic data, and self-report questionnaires. Each team undergoes up to six consecutive phases: three behavioral tasks, one mission training session, and two collaborative SAR missions. This dataset will support studying a large variety of research questions on topics including teamwork, coordination, plan recognition, affective computing, physiological linkage, entrainment, and dialog understanding. We provide an initial public release of the de-identified data, along with analyses illustrating the utility of this dataset to both computer scientists and social scientists.

## 1   Introduction

Teams of the future will increasingly involve humans and AI agents working together as trusted partners, leveraging their complementary skills to achieve shared goals. The efficacy of AI teammates will be enhanced if they are able to understand the beliefs, desires, and intentions of their human teammates, i.e., if they have a *machine theory of mind* (MToM) [1, 2]. However, this capability alone is not sufficient—they will need to understand the interpersonal dynamics *between* their human teammates, that is, they will need a *machine theory of teams*. A natural first step for constructing a (computational) theory of teams is to draw upon the vast amount of existing literature on teamwork in purely human teams. However, as Roberts et al. [3] note, significant additional work is needed to extend existing models of teamwork to human-machine teams.

We present the ToMCAT (Theory of Mind-based Cognitive Architecture for Teams) dataset—a rich, multimodal dataset developed to significantly advance our understanding of teaming in both purely human and hybrid human-machine teams. The dataset contains data from experiments in which teams of three humans (and optionally, an AI advisor) execute complex collaborative tasks—specifically,

urban search-and-rescue (USAR) missions—in a virtual Minecraft-based testbed [4]. However, we emphasize that the primary focus of this paper is not on the USAR missions themselves, but rather the complex social behaviors that the testbed is designed to elicit. Additionally, we instrument the participants with additional sensors that capture data via the following modalities: spoken communications, gaze, facial image captures, galvanic skin response (GSR), electroencephalography (EEG), and functional near-infrared spectroscopy (fNIRS), and have them perform a set of novel behavioral baseline tasks (§ 4).

**Main contributions. 1)** A rich multi-person, multimodal dataset for teams collaborating on complex, time-constrained tasks, **2)** Three structured behavioral baseline tasks designed to ground the physiological data, as well as serve as independent multi-person, multi-task, multimodal datasets; and **3)** Exploratory analyses suggestive of the huge space of inquiry possible with this data.

## 2 Related work

This paper builds upon multiple lines of research related to human-machine teaming, synthetic task environments, and interpersonal coordination as evidenced by physiological linkage. The primary motivation for this dataset is to accelerate the development of effective artificial agent teammates with artificial social intelligence through a deeper understanding of purely human and human-machine teaming in cognitively complex, time-constrained scenarios.

**Machine Theory of Mind**   Given the core role played by theory of mind (ToM) [5] in human social intelligence, there has been steadily increasing interest in developing artificial agents with MToM [1, 2, 6–22]. A significant portion of the literature on MToM, however, evaluates it in contexts that are either disembodied (e.g., image classification and purely language-based tasks [6, 12, 16]) or 'lightly-embodied' (e.g., text adventure games [13] and small 2D gridworlds [1, 2, 14, 18]). Voxel-based environments such as Minecraft [23] represent a natural step up in complexity.

**Minecraft for AI research**   Minecraft is an open-world adventure game that is gaining popularity as an AI testbed due to its ability to support diverse tasks [24], modifiability, and large user base. Of the projects that use Minecraft for AI research, a large number use it as a testbed for reinforcement learning [25–27], or for training AI agents to perform tasks based on natural language commands [28–32]. However, following others [8, 9, 11, 15, 21, 33–39], we are interested in using Minecraft as a testbed for ToM and human-machine teaming—that is, rather than having AI agents execute tasks within a Minecraft-based environment, we have *humans* executing the tasks, with AI agents (if present) acting as passive observers or advisors.

**USAR tasks in synthetic task environments**   The environment we use [4, 40] is an example of a *synthetic task environment* (STE) [41]—a medium-fidelity simulation environment that strikes a pragmatic balance between highly abstracted lab settings (more controlled, cheap, less likely to generalize to real-world situations) and high-fidelity simulations/real-world environments (less controlled, expensive, more representative of real-world scenarios). The USAR task was chosen due to its time-constrained, cognitively demanding nature, coupled with the potential for humans and robots to perform complementary team roles in real-world USAR scenarios [33, 42]. The use of Minecraft-based USAR STEs to study human-machine teaming is relatively well-established—they have been used for small-scale studies [33, 43] as well as large-scale datasets [38, 39] upon which numerous analyses have been performed [8, 10, 11, 15, 17, 21, 44].

The dataset most closely related to ours is the ASIST Study 3 dataset [39]. We use the same STE [4] and Minecraft USAR tasks [45]. However, our data differs from theirs in several ways, the most important of which is the inclusion of a number of of additional sensing modalities in our experiments: gaze, facial image capture, EKG, GSR, EEG, and fNIRS (see § 2.1 for more details), which opens up a number of additional research opportunities.

First, since we are concerned with theory of mind, recording EEG and fNIRS signals provides us a way to ground inferences about cognitive and affective components of human mental state in rich data that reflects the actual underlying brain activity of the participants—in a sense, getting us closer to the 'ground truth'. Notably, the study of the affective component of human mental states (i.e., emotions) is conspicuously absent from existing works on MToM, despite affect playing a crucial role in human social interactions and decision making. Additionally, affect is reflected in other forms,

e.g. facial expressions and changes in heart rate, neither of which are represented in the ASIST Study 3 dataset. While we do not expect, say, USAR team members to wear fNIRS/EEG caps in the field, we believe that the modalities in this dataset will allow us to create mappings between surface-level indicators of affect and coordination (e.g., facial expressions, tone of voice) and deeper underlying affect and coordination as detected from fNIRS/EEG signals.

Second, physiological linkage (PL)—i.e., statistical association between the physiological markers of two or more people over time [46]—has been shown to be predictive of performance and attributes [47, 48]. Furthermore, PL may depend on the interaction of individual differences and context—e.g., differences in individual social skills and attachment styles have been found to be associated with qualitatively distinct forms of PL in competitive and collaborative contexts [49]. Therefore, PL may be a promising predictor and outcome of team dynamics. Through the addition of physiological sensing modalities, our dataset enables the study of PL in the context of human-machine teams.

Third, unlike in ASIST Study 3, the participants in our study perform a set of tasks designed to compare physiological changes and phonetic entrainment resulting from performing team tasks. Finally, instead of running multiple AI advisor experimental conditions, we only have one (the ToMCAT agent), resulting in a much larger amount of data for this single advisor, which will enable increased statistical power for analyses that are not focused on comparing the outcomes of interventions by different AI advisors (one of the primary goals of ASIST Study 3).

**Open-access fNIRS datasets**  Notably, to the best of our knowledge, our dataset is also the **largest** open-access fNIRS dataset to date. It is approximately **13.5 times** the size of the fNIRS2MW [50] dataset (1.5× more subjects and ≈ 9× more fNIRS data per subject).

## 2.1  Physiological measures

We simultaneously record multiple subjects' neural activities (hyperscanning) during real-time social interactions [51]. Based on previous literature on team cognition and hyperscanning [52], we selected EEG, fNIRS, EKG, GSR, and gaze as our main physiological measurement modalities. These were collected for each participant for the duration of the group session (see § 3). This approach provides opportunities for data analysis at various time resolutions [52], optimal variable control by using data from different modalities for denoising, and higher-level feature selection/construction [53]. Details on the equipment and procedures for data acquisition and signal processing are provided in the appendices.

**EEG**  EEG is a non-invasive measure of the scalp electrical activity generated from the cerebral cortex. It provides data on brain activity with temporal resolution on the order of a millisecond [54]. This high temporal resolution affords us opportunities for event-related (task-based) analysis (event-related potential, ERP), which has been widely adopted in cognitive science research on various topics including decision-making, emotion elicitation, and team cognition [55]. EEG hyperscanning has yielded fruitful results for human-human social interaction research [52]. Sinha et al. [51] found that inter-brain synchrony calculated from simultaneous EEG recordings of paired subjects was found to be significantly higher when the subjects were in a cooperative scenario compared to when they were in a competitive scenario. EEG has also been used to study human-machine teaming scenarios—e.g., Shayesteh, Ojha, and Jebelli [56] measured EEG signals from subjects performing a collaborative construction task with a virtual robot in an immersive environment, and found that a $k$-nearest neighbors model (kNN) trained on EEG signals was able to predict the human's level of trust in the robot with an accuracy of ≈88%.

**fNIRS**  fNIRS is an non-invasive optical brain imaging technology that assesses the contrast between oxygenated and de-oxygenated hemoglobin in the cortex, and uses the hemodynamic fluctuation as an indirect measure of brain activity in targeted brain areas [52]. While fNIRS has a lower temporal resolution than EEG [57] (on the order of 10 ms), it is highly portable and less susceptible to motion, making it an increasingly popular modality for social interaction experimental settings [58]. For this study, we use optodes that mainly cover the frontal lobe area, based on previous research [52, 59, 60] that found that greater interpersonal brain synchronization occurs at the frontopolar area, indicating better coordination performance. Oxygenation changes in the prefrontal cortex (PFC) have also found to be related to performance on various individual cognitive tasks [61], including language translation and switching [62], verbal fluency [63], and mental manipulation [64].

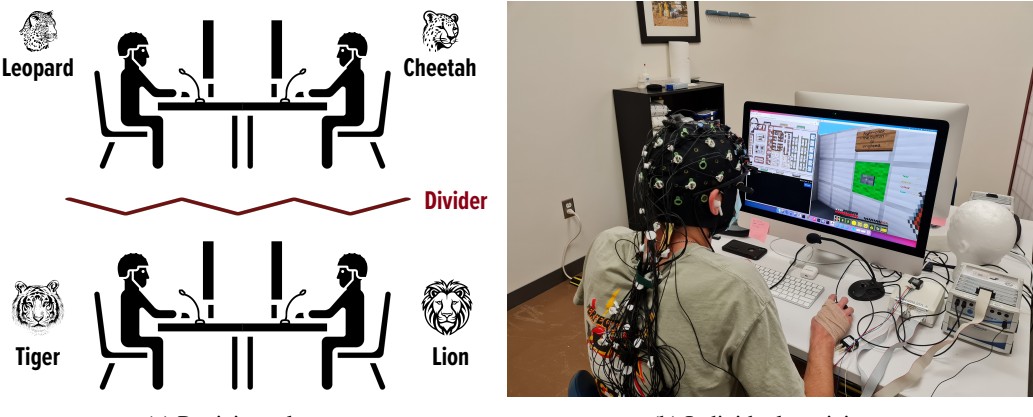

| (a) Participant layout | (b) Individual participant |

Figure 1: Our experimental setup for data collection. Figure 1a shows the layout of the participants. The 'Lion', 'Tiger', and 'Leopard' stations are for regular participants, while the 'Cheetah' station is used for the experimenter that joins the group session for the competitive ping-pong task (§ 4.3). The 'Lion' and 'Tiger' stations are separated from the 'Leopard' and 'Cheetah' stations by a divider, in order to reduce audio cross-contamination between the participants' microphones. Figure 1b shows a more detailed view of an individual participant, who is instrumented to record EEG, fNIRS, GSR, EKG, and gaze data and in the midst of a Minecraft SAR mission.

**EKG**  An electrocardiogram (EKG) measures heart activity over time and offers high temporal resolution. Common EKG signal derivatives include inter-beat interval (i.e., the time between heartbeats) and respiratory sinus arrhythmia (i.e., variability in heart rate due to breathing). These signals result from coordinated biological activity within a person and are commonly used to model coordination between people, as reflected by physiological linkage. Additionally, EKG data can be used to filter out systematic cardiac activity noise (1–1.5 Hz [65]) from fNIRS data.

**GSR**  Galvanic skin response (GSR), is a measure of electrical conductivity on the surface of the skin [66]. Sweat gland activity varies unconsciously and automatically, peaking approximately 1–5 seconds after stimulus onset. GSR is commonly used in the study of emotion processes and teamwork. For example, studies have found that GSR activity is associated with team performance [67], mental effort [68], and self-reported emotion during team tasks [69]. For our study, we were interested in the peak amplitudes (i.e., change from stimulus onset to highest peak), which can be used to examine sweat gland activity within and between teammates following a stimulus presentation or during team-based tasks.

**Gaze**  An eye tracker works by shining infrared light onto the eyes of a participant, creating reflections on the corneae that are then used to identify the locations of their pupils. By capturing their eye/pupil movements, the eye tracker software can infer the point of gaze (where the participant is looking) in real time [70, 71]. Eye-tracking is widely used in a variety of disciplines, including psychology [70], marketing [72], and UI/UX research [73]. Additionally, eye-tracking can provide event markers for other modalities such as EEG, fNIRS, and EKG.

## 3   Experimental design

The study was held at the University of Arizona. Individuals were deemed eligible if they were at least 18 years of age, read and spoke English, and did not have any major physical limitations that would interfere with completing tasks on a computer. Interested individuals contacted the research team via e-mail, text, or phone. Details on the ethical review and the recruitment process are provided in § A.3 and Appendix B respectively. All participants were compensated with either an Amazon gift card or course credit. Participants were asked to complete a 30-minute individual 'pre-session' and a 3-hour 'group session'.

### 3.1 Pre-session

Participants started by completing the online consent form if they had not already done so. We then measured their heads to select an appropriately sized EEG/fNIRS cap for them.

**Speech elicitation tasks**  We conducted two speech elicitation tasks, collecting speech data for each participant prior to their interaction with their teammates. This created a speech baseline for each player prior to their interactions with their teammates and was used to study phonetic entrainment between teammates during the course of the Minecraft missions. For more details, see Appendix K. Entrainment is a useful method for assessing the dynamics of a social interaction and levels of rapport [74, 75]. Prior work has found strong correlations between entrainment and success in group tasks [74, 76, 77].

**Questionnaires**  The following questionnaires were administered: (i) a COVID-19 health screener, (ii) a survey that collected information about basic demographics (e.g., sex, racial background, household income, highest level of education), experience with video games, and health (e.g., speech/hearing, language, impairments, diagnoses, psychoactive medication, etc.), (iii) the Big Five Inventory- 2 Short Form (BFI-2-SF) personality questionnaire [78], and (iv) the Attachment Style Questionnaire that assessed adult attachment [79].

### 3.2 Group session

When required, the participants interacted with each other through a keyboard and mouse located at each experimental station. Each participant was positioned in front of a computer monitor, with dividers used to increase physical separation between the participants (see Figure 1). If only two out of the three planned participants showed up to the group session, or if one of the participants dropped out in the midst of it, a confederate (i.e., a member of our research team) would step in to take their place. Out of the 1014 task instances for which data was supposed to be recorded for regular participants, 147 (i.e., 14.5%) had experimenters filling in for participants.

**Baseline tasks**  Participants started by conducting a set of behavioral baseline tasks (§ 4).

**Search-and-rescue missions**  Next, participants conducted the Minecraft-based SAR missions described by Huang et al. [45]. Each team conducted a 20-minute tutorial mission, followed by two 17-minute main missions: Saturn A and Saturn B. The tutorial mission consists of a series of tasks designed to familiarize participants with the game environment and their avatars' specific abilities. The first 2 minutes of the main missions were devoted to planning—participants were encouraged to discuss strategies and review good and bad practices they adopted in the previous mission. In the next 15 minutes, access to the building is unblocked and participants can effectively start to earn points, which happens after they find, treat and move victims of an in-game building collapse to assigned safe areas. A subset of the teams were advised by the ToMCAT AI agent [80], which was designed to improve team coordination by intervening on team communication.

**Post-game survey**  After completing the Minecraft tasks, participants completed a brief post-game survey. They were asked to rate their emotions due to (i) ther AI agent teammate, (ii) how the game went, and (iii) the other team members, on a scale of 'not at all' to 'a very large amount'. In addition, participants indicated their impression of the agent and other team members by sliding a bar between a pair of adjectives (e.g., intelligent–unintelligent, inexpert–expert, etc.). Lastly, participants responded on the extent to which they disagreed or agreed to general statements along with statements about other team members and the agent (e.g., "It seemed like my emotional reaction was wrong or incorrect because of the agent's response.").

## 4  Baseline tasks

We collected rest state physiological data and conducted three behavioral baseline tasks to ground complex physiological signals expected in collaborative missions to simpler, well-studied settings. For example, we can map patterns in the Minecraft tasks to patterns of coordination established in a simple task. These tasks closely resemble well-established ones, facilitating connections with

existing research. Our approach extends prior studies by (i) addressing the underexplored area of emotion in hyperscanning [81] through affective behavioral tasks and (ii) expanding from the prevalent two-person paradigm to experiments involving three participants.

## 4.1 Rest state and finger tapping tasks

**Rest state task** This task was designed to collect baseline physiological data from the participants while they were in a resting state. This is consistent with standard neurophysiological research [82] and important for comparison purposes—by establishing a resting baseline, we can better understand an individual's functional baseline in the absence of exposure to the stimuli in the other tasks. In this task, participants sat quietly for 5 minutes without engaging in any activity, with their monitor displaying a countdown timer showing them the time remaining in the task.

**Finger-tapping task** This task was designed to record physiological data during team synchronization in a cooperative activity. It allows us to observe neural processes during synchronization. Results from Tognoli et al. [83] suggest certain neural correlates when participants are tasked with finger-tapping with and without visual cues from each other. Furthermore, hyperscanning [52] studies have shown an association between EEG signals and behaviors [81, 84]. Our finger-tapping task is a variant of the one proposed by Tognoli et al. [83].

## 4.2 Affective task

In this task, participants viewed a curated set of images (see Appendix L) designed to elicit various emotions. We employed Russell's valence-arousal scale [85]—a widely recognized tool in affective research—to quantitatively assess emotions. The task aims to collect physiological and emotional data for interpreting emotional experiences based on physiological responses in subsequent tasks. Prior studies indicate a connection between fNIRS, EEG, and autonomic functioning during the processing of emotions—specifically, PFC activation [86] and potential dual motive systems in the brain [87]. These findings align with literature on the PFC's role in memory, emotion regulation, and cognition [88–92]. The affective task includes 'individual' and 'team' affective subtasks.

**Individual Affective Task** For each image, the participant was shown the following sequence: (i) a black screen for one second, (ii) a '+' icon at the center of the screen (guiding the participant's attention to the center) for 0.5 seconds, (iii) the image itself for 5 seconds. After viewing each image, participants had 20 seconds to rate the emotions they experienced during their observation. The rating screen presented a 5-point valence scale (-2 for *upset* to +2 for *happy*) and a 5-point arousal scale (-2 for 'calm' to +2 for 'excited'), with 0 denoting 'neutral'. These scales were adapted from the Self-Assessment Manikin (SAM) [93] pictorial rating—a non-verbal technique for gauging individual affect. Participants were prompted to register their emotional responses and submit them before the onset of the subsequent image.

**Team Affective Task** This task is similar to the individual affective task, except that participants viewed each image together instead of separately, after which they discussed their emotional experience and submitted a single rating representing their collective emotional experience in response to the image.

## 4.3 Ping-pong task

The primary objective of this task is to collect neurophysiological data when participants are engaged in competitive and cooperative scenarios. A range of tasks including card games, ping-pong, and music and rhythm synchronization exercises have been used in the hyperscanning literature—e.g., studies have found evidence of inter-brain synchronization among participants collectively perform a piece of music [94, 95] and in a cooperative task based on the Prisoner's Dilemma [96]. The ping-pong task is divided into two subtasks: competitive and cooperative.

**Competitive Ping Pong Task** Inspired by Sinha et al. [97], our competitive ping-pong task has participants compete against each other in a 2-minute 1-on-1 computer-based ping-pong game. Typically, two of the three participants would compete against each other, while the third competed against a confederate. Players controlled an on-screen paddle with a mouse. Paddles were positioned

on the left and right sides of the screen, and constrained to move solely vertically. Participants scored a point whenever the ball hit the wall of the side opposite to that of their paddle. After hitting the wall, the ball would ricochet back with the vertical component of its velocity being randomized. Prior to the match, a 10-second familiarization phase was provided during which the participants could practice moving their paddles while the ball remained stationary at the center of the screen.

**Cooperative Ping Pong Task**   This task was similar to the competitive version, except that all three participants were on the same side, playing against an AI agent instead of against each other. The horizontal component of the ball's velocity during the cooperative task was higher than that in the competitive task. The participants' paddles could move through each other and were on the left side of the screen, while the AI agent's paddle was on the right. Similar to the competitive task, a 10-second familiarization phase was provided before the start of the match.

## 5   Exploratory experiments

We developed two simple experiments to illustrate the large scope of new studies our data set can support. The first is designed to compare the power of EEG and fNIRS data to predict self-reported affect, and the second explores whether synchronization of EEG and fNIRS signals among team members is predictive of team performance. Note that while we do not fuse the data from the two modalities, these experiments are intrinsically multimodal since the EEG and fNIRS modalities are recorded simultaneously for each participant, thus enabling us to compare their predictive power.

We emphasize that these experiments are *exploratory* (i.e., meant to recognize novel patterns in the data, which can potentially lead to the generation of new hypotheses) rather than *confirmatory* (i.e., testing existing hypotheses). Both types of experiments are required for scientific progress [98]. Exploratory experiments are especially appropriate when the topic of research—in our case, machine learning based on brain data—is relatively less well-studied compared to other modalities (due to the inherently challenging nature of collecting brain data) such as images, text, and audio.

### 5.1   Predicting affect from brain scan data

Predicting affect from brain scan data is a challenging, yet intriguing endeavor in the realm of neuroscience and affective computing. In this study, we use a multimodal dataset comprising EEG and fNIRS data to examine the feasibility of predicting individuals' self-reported valence and arousal using data from the individual affective task (§ 4.2). We focus on specific regions of the head to ensure spatial alignment of EEG and fNIRS data, enabling the comparison of these two modalities.

Most work on mapping brain scan data to valence and arousal uses EEG data. This includes Rayatdoost et al. [99], who developed a deep domain adversarial neural network (DANN) to link EEG data to valence and arousal, achieving average classification accuracies of 72.8% and 65.0% for valence and arousal respectively on the MAHNOB-HCI database [100], and accuracies of 69.8% and 57.6% for valence and arousal classification on the DAI-EF database [101]. Galvão, Alarcão, and Fonseca [102] used kNN on features derived from the EEG frequency domain to predict valence and arousal values, achieving accuracies of 79.4%, 83%, and 80.6% on the DEAP [103], AMIGOS [104], and DREAMER [105] datasets, respectively. For all five datasets, participants watched videos and subsequently rated their emotions using SAM scales [106] for valence and arousal (a 9-point scale for MAHNOB-HCI, DAI-EF, DEAP, and AMIGOS, and a 5-point scale for DREAMER)

Bandara et al. [107] used a support vector machine (SVM) to predict valence and arousal scores from fNIRS data, achieving an $F_1$ score of 0.74 on the DEAP dataset. For images, Trambaiolli, Biazoli, Cravo, et al. [108] achieved a classification accuracy of 89% in discerning positive from negative valence using fNIRS signals and a linear discriminant analysis (LDA). Finally, Sun, Ayaz, and Akansu [109] combined EEG and fNIRS data, and used an SVM to classify with 75% accuracy.

While many studies use a SAM scale with ranges of 1–9, 1–5, or 1–10, we use a scale with a range of −2 to +2 for a more compact scale and a clear neutral reference. We use brain scan data obtained from EEG and fNIRS recording trimmed to sets whose EEG and fNIRS samples overlap as best as possible. Specifically, for EEG we selected channels FC5, FCz, FC6, F7, F8, AFF1h, and AFF2h, whereas for fNIRS we selected channels Fz-F1, Fz-F2, F3-F7, F3-F1, F4-F2, F4-F6, AF3-F7, AF3-Afz, AF4-F6, and AF4-Afz. We trained separate CNNs [110] for fNIRS and EEG data. The fNIRS data were kept

Table 1: Accuracy and loss (mean ± standard error of the mean, computed over 5 folds) for classification of valence and arousal scores.

| Model | Offset (s) | Window size (s) | Accuracy (%) | | Loss |
|---|---|---|---|---|---|
| | | | Valence | Arousal | |
| CNN$_{fNIRS}$ | 0 | 2 | 26.8 ± 1.2 | 30.1 ± 0.8 | 3.02 ± 0.012 |
| CNN$_{fNIRS}$ | 2 | 2 | 30.4 ± 1.2 | 28.0 ± 0.8 | 3.02 ± 0.008 |
| CNN$_{fNIRS}$ | 5 | 2 | 28.4 ± 1.2 | 30.3 ± 0.8 | 3.01 ± 0.007 |
| CNN$_{EEG}$ | 0 | 1 | 29.3 ± 1.4 | 29.8 ± 1.3 | 2.80 ± 0.004 |
| Baseline$_{fNIRS}$ | N/A | N/A | 29.0 ± 0.9 | 29.7 ± 0.1 | N/A |
| Baseline$_{EEG}$ | N/A | N/A | 27.5 ± 0.7 | 30.3 ± 0.5 | N/A |

in the spatial domain, while EEG data were segmented into frequency bands (theta, alpha, beta, and gamma) and their wavelet features were extracted with four levels of decomposition.

fNIRS signals exhibit a phenomenon known as the hemodynamic response factor (HRF) [111, 112] which represents the relationship between neural activity and the corresponding changes in blood oxygenation levels that occur in response to that activity (when a participant views a specific image). The HRF consists of two phases: (i) the *initial dip*, which typically lasts for 1–2s and involves an initial drop in the concentration of oxygenated hemoglobin (HbO) and a simultaneous increase in deoxygenated hemoglobin (HbR) shortly after neural activation, and (ii) the *hemodynamic response peak*, in which there is an increase in HbO concentration and a decrease in HbR concentration, resulting in a peak in HbO concentration that occurs ≈ 4–6s after activation.

We present our results in Table 1. We tried several offsets for the fNIRS data to see if there was a noticeable effect due to the HRF as discussed above. We note that accuracies for EEG and fNIRS are not precisely comparable due to the disparity in participant numbers—97 for EEG and 102 for fNIRS. Unfortunately, our attempt at using basic CNNs for classification did not perform any better than the baseline. In contrast to prior work, we held out data in units of participant-image pairs, rather than predicting solely within participants. The relationship between functional brain regions and cap location varies among participants. While the training data did contain some data for held out participants looking at *other* images, most of the training data was for other participants. Success on this task will likely require addressing functional regions to individualized cap locations, as well as more effort on neural network design. Given basically baseline performance, we are not surprised that accounting for the HRF with an offset did not make any real difference. Further details and quantitative results can be found in the confusion matrices provided in § F.1.

## 5.2 Linking temporal correlation of brain signals with scores

Shared cognition in team environments is gaining interest, particularly in understanding social dynamics that lead to successful performance, and in developing intervention techniques to enhance collaboration [113, 114]. Research has consistently shown that cooperation plays a vital role in influencing overall task performance [113], enabling coordination and information sharing, which enhances the effectiveness of the team [115, 116]. Studies have found behavioral and neurological synchronization between subjects during cooperative tasks [115–118], reinforcing the idea that cooperation may go beyond mere action coordination or knowledge sharing and may suggest a shared mental model within cooperative settings that includes coordination and sharing of social content like emotion and intentions [113, 119].

Research efforts on these fronts will benefit from more comprehensive data, specifically, data containing a larger set of modalities, and from multiple interacting participants. Existing studies mostly focus on limited brain regions with a single modality (i.e., either fNIRS or EEG but not both) during a single task [116, 117], which undermines their ability to capture the complex nature of shared mental models. Our research extends previous methods of classifying cognitive processes in single-participant studies [120] to identifying cognitive processes in teams, thus enabling the study of shared mental models of teams.

In this second experiment to illustrate the potential of our dataset, we study whether synchronization of EEG and fNIRS signals between team members can predict team performance. Building on research establishing a connection between synchronization and higher cooperation levels [115–117], we examine the brain holistically for associations between the correlation of EEG and fNIRS data

Table 2: Linear regression coefficient $\beta$ (slope) and associated significance ($p$-value), cross-validation with leave-one-out mean error ($\bar{e}$) with standard error of those means ($\sigma_{\bar{e}}$) in parentheses for each fNIRS and EEG channel over average correlation from all experiments. fNIRS (EEG) channels with $R^2 < 0.165(0.075)$ for all three tasks are excluded from the table. Blue and red numbers indicate positive and negative slopes respectively. $R^2$ of the null model for all channels is 0.0. The values under the $\bar{e}^*$ and $\beta^*$ columns are in units of $10^2$. $p$-values less than 0.05 are in **bold**.

| Channel | Ping Pong Cooperative | | | | Minecraft Saturn A | | | | Minecraft Saturn B | | | |
|---|---|---|---|---|---|---|---|---|---|---|---|---|
| | $R^2$ | $p$ | $\bar{e}\,(\sigma_{\bar{e}})$ | $\beta$ | $R^2$ | $p$ | $\bar{e}^*\,(\sigma_{\bar{e}})$ | $\beta^*$ | $R^2$ | $p$ | $\bar{e}^*\,(\sigma_{\bar{e}})$ | $\beta^*$ |
| fNIRS | | | | | | | | | | | | |
| F3-F5 (HbO) | 0.03 | 0.30 | 2.3 (.3) | 2.45 | 0.10 | 0.07 | 1.5 (.2) | −2.99 | 0.27 | **0.01** | 1.1 (.1) | −7.60 |
| F3-F1 (HbO) | 0.00 | 0.89 | 2.3 (.3) | 0.43 | 0.07 | 0.12 | 1.6 (.2) | −2.40 | 0.17 | **0.03** | 1.2 (.2) | −9.13 |
| Fz-AFz (HbO) | 0.02 | 0.35 | 2.2 (.3) | 2.11 | 0.23 | **0.00** | 1.3 (.2) | 4.56 | 0.02 | 0.44 | 1.2 (.2) | −2.73 |
| FPz-FP1 (HbO) | 0.19 | **0.01** | 2.1 (.3) | 6.43 | 0.09 | 0.08 | 1.6 (.2) | −2.64 | 0.01 | 0.62 | 1.3 (.2) | −2.66 |
| FPz-AFz (HbO) | 0.00 | 0.68 | 2.3 (.3) | 0.77 | 0.26 | **0.00** | 1.3 (.2) | 4.49 | 0.02 | 0.45 | 1.2 (.2) | −3.45 |
| F4-F2 (HbO) | 0.00 | 0.74 | 2.3 (.3) | −0.77 | 0.01 | 0.50 | 1.7 (.3) | 1.02 | 0.20 | **0.02** | 1.2 (.2) | −10.32 |
| AF3-FP1 (HbR) | 0.11 | 0.04 | 2.2 (.3) | 3.90 | 0.19 | **0.01** | 1.4 (.2) | −3.94 | 0.05 | 0.27 | 1.2 (.2) | −3.41 |
| FPz-FP2 (HbR) | 0.00 | 0.78 | 2.3 (.3) | −0.58 | 0.21 | **0.00** | 1.3 (.2) | −4.32 | 0.01 | 0.56 | 1.2 (.2) | 2.16 |
| AF8-F6 (HbR) | 0.00 | 0.94 | 2.4 (.4) | 0.17 | 0.29 | **0.00** | 1.3 (.2) | −4.99 | 0.10 | 0.10 | 1.2 (.2) | 4.13 |
| EEG | | | | | | | | | | | | |
| FCz | 0.09 | 0.08 | 2.3 (.3) | −4.20 | 0.27 | **0.00** | 1.2 (.2) | −3.66 | 0.12 | 0.08 | 1.2 (.2) | −2.77 |
| AFF1h | 0.14 | **0.02** | 2.3 (.3) | −5.79 | 0.00 | 0.75 | 1.6 (.2) | 0.60 | 0.02 | 0.53 | 1.3 (.2) | 0.79 |
| O2 | 0.04 | 0.24 | 2.4 (.3) | −2.54 | 0.24 | **0.00** | 1.4 (.2) | 4.44 | 0.02 | 0.54 | 1.3 (.2) | 0.88 |
| TP9 | 0.13 | **0.03** | 2.3 (.3) | −6.98 | 0.02 | 0.43 | 1.6 (.2) | −1.06 | 0.17 | **0.04** | 1.2 (.2) | 3.88 |
| P3 | 0.00 | 0.70 | 2.4 (.3) | −0.90 | 0.00 | 0.76 | 1.6 (.2) | 0.56 | 0.10 | 0.11 | 1.2 (.2) | 1.81 |
| P8 | 0.06 | 0.16 | 2.3 (.3) | −3.30 | 0.25 | **0.00** | 1.2 (.2) | −4.53 | 0.00 | 0.80 | 1.3 (.2) | −0.51 |
| C4 | 0.00 | 0.91 | 2.4 (.4) | −0.31 | 0.07 | 0.13 | 1.5 (.2) | 3.03 | 0.09 | 0.14 | 1.3 (.1) | −4.56 |
| AFF2h | 0.03 | 0.32 | 2.4 (.3) | −2.17 | 0.17 | **0.02** | 1.4 (.2) | 4.45 | 0.02 | 0.54 | 1.3 (.2) | −0.98 |
| T8 | 0.03 | 0.33 | 2.4 (.3) | −2.29 | 0.00 | 0.79 | 1.6 (.2) | −0.47 | 0.08 | 0.18 | 1.3 (.2) | −1.88 |

channels between participants with scores on the ping-pong tasks (a more constrained laboratory setting), and the two Minecraft-based SAR missions (a more naturalistic setting). We expect that stronger linkage between certain parts of the brain will be associated with better performance. Our analysis methods are similar to those in existing studies [116, 117]. Specifically, we are looking at whether the temporal correlation across participant brain signals can predict team performance.

For each of the three tasks, for each of the EEG and fNIRS channels, we computed the Pearson correlation of the brain data between the three participants in a team by taking the average of their pairwise correlations. If there were only two participants, we simply used their correlation. We compute correlations by synchronizing the measurements to a common start time and frequency (500 Hz for EEG and 10 Hz for fNIRS) using linear interpolation. We then used the correlation for each EEG and fNIRS channel as the predictor for the final task performance score as the outcome variable in a linear regression model fit to these variables over all teams. We evaluated the performance of the resulting models using the $R^2$ measure, which quantifies the proportion of variance in the task performance that can be accounted for by the EEG and fNIRS correlations, and cross-validation with leave-one-out mean error, where we fit the regression model over all but one team and evaluate prediction error on the one held-out team in each fold. Our results are shown in Table 2.

For the Ping Pong Cooperative baseline task, many fNIRS channels had a positive slope for the fit of the task performance predicted by fNIRS synchronization. Conversely, the linkage between EEG-measured brain regions among participants and task performance often had negative slopes. In our Minecraft Saturn A mission, participants were in the process of familiarizing themselves with the task's challenges. Here, the EEG/fNIRS correlation between participants and task performance was varied, without a clear pattern. However, in the Minecraft Saturn B mission, in which participants were now familiar with the mission and each other, there was a larger number of fNIRS and EEG channels with a negative correlation between brain data linkage and task performance when compared to Minecraft Saturn A. Understanding what is going on will require more thorough analysis on where we suspect spurious results, mapping to brain function as seen in the baseline tasks, and accounting for the differences in what we expect EEG and fNIRS signals to tell us.

# 6 Dataset usage

**Accessing the dataset**   The dataset and its documentation is available at https://tomcat.ivilab. org. We provide access to the data in two ways. The first is through a Datasette [121] instance, which provides graphical and programmatic interfaces for users to explore the data, retrieve subsets of it that they are interested in, or simply download the backing SQLite database. The second is in the form of pre-built files containing subsets of data that we (i) used for the experiments in this paper, and (ii) expect will be commonly requested by other researchers.

A key issue with using the raw data is that it comprises data from multiple asynchronous data streams. Hence, aligning the data for multiple participants entails interpolation. A second issue is recording specific issues encountered during data collection, such as an experimenter stepping in for a participant. Finally, derived data will also include standard data transformations and cleaning.

**Continued engagement**   The scale and complexity of this dataset make it infeasible for us to annotate every type of label that might be of interest. Rather, we hope that the release of the dataset will seed the development of a community that works together to fully explore this data. We envision a process by which researchers build upon this dataset by adding layers of annotations for labels of interest, unlocking the ability to answer additional research questions. We are also happy to include pointers on our website to papers that use this dataset, in order to facilitate connections between researchers working with this dataset.

We *highly* encourage users to sign up for our mailing list to get updates on data issues and annotation layers, benchmarks, and documentation.

## 6.1 Additional usage examples

In § 5, we presented illustrative experiments in that study the affective component of ToM and the relationship between brain signal correlations and team performance on a collaborative task. One could study other aspects of ToM as well.

**Intention detection**   For example, *intentions* are commonly considered part of human mental states. Consider a researcher who is interested in developing algorithms to infer participant intentions from observed behavior (i.e., *plan recognition* [122]). This could be done by encoding the observations as sequences of discrete actions—either manually or semi-automatically, depending on the temporal, spatial, and semantic granularity of interest. Data that could be used for this include the participant's in-game position and velocity, as well as the semantic contents of their utterances.

**Utterance classification**   Another example use case involves labeling participant utterances with sentiment, emotion, and dialog act labels will enable developing models for sentiment, emotion and dialog act classification in task-related dialog. Unlike previous dialog datasets, the ToMCAT dataset contains text, speech, and physiological data that can be simultaneously leveraged for these classification tasks.

# 7 Conclusion

In this transdisciplinary work, we integrate numerous threads of research in computer science, psychology, and cognitive science to present the ToMCAT dataset, which to our knowledge is the only dataset that contains both (i) data on human-machine teaming in a complex synthetic task environment and (ii) rich physiological data from a number of sensors, including fNIRS and EEG, thus enabling the exploration of fundamental research questions related to the interplay of competition, cooperation, and neurophysiological responses in human-machine teams. Furthermore, to our knowledge, this is the largest open-access fNIRS dataset currently available. Finally, we conduct exploratory experiments linking valence, arousal, and team performance to brain signals, illustrating the potential of the dataset for exploring a variety of research questions. We are excited to share this dataset with the community and look forward to seeing the research findings it enables.

## Acknowledgments and Disclosure of Funding

Research was sponsored by the Army Research Office and was accomplished under Grant Number W911NF-20-1-0002. The views and conclusions contained in this document are those of the authors and should not be interpreted as representing the official policies, either expressed or implied, of the Army Research Office or the U.S. Government. The U.S. Government is authorized to reproduce and distribute reprints for Government purposes notwithstanding any copyright notation herein. We would also like to acknowledge intramural funding from the University of Arizona's SensorLab.

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

# Appendix

## Table of Contents

# A Datasheet

## A.1 Motivation

*For what purpose was the dataset created?* Was there a specific task in mind? Was there a specific gap that needed to be filled? Please provide a description.

The dataset was created to significantly advance our understanding of teaming in both purely human and hybrid human-machine teams. While there exist prior datasets involving human-AI collaboration in complex open-ended environments, none include the large number of physiological measures that we provide.

*Who created the dataset (e.g., which team, research group) and on behalf of which entity (e.g., company, institution, organization)?*

This dataset was created by members of the ToMCAT research project (`https://ml4ai.github.io/tomcat`), spanning five departments (School of Information, Computer Science, Norton School of Human Ecology, Psychology, and Linguistics) and three labs (ML4AI, IVILab, and the TIES Lab) at the University of Arizona.

*Who funded the creation of the dataset?* If there is an associated grant, please provide the name of the grantor and the grant name and number.

The creation of this dataset was funded by the Army Research Office and was accomplished under Grant Number W911NF-20-1-0002. The grant was awarded through the Defense Advanced Research Projects Agency (DARPA). We would also like to acknowledge intramural funding from the University of Arizona's SensorLab.

## A.2 Composition

*What do the instances that comprise the dataset represent (e.g., documents, photos, people, countries)?* Are there multiple types of instances (e.g., movies, users, and ratings; people and interactions between them; nodes and edges)? Please provide a description.

The dataset consists of data from experiments in which teams of three humans (and optionally, an AI advisor) execute urban search-and-rescue (USAR) missions in the ASIST Study 3 testbed [4], which is based on Minecraft and instrumented to capture in-game states, actions, and events with a high level of granularity. Additionally, we instrument the participants with additional sensors that capture data via the following modalities: spoken communications, eye tracking, facial image captures, galvanic skin response (GSR), electroencephalography (EEG), and functional near-infrared spectroscopy (fNIRS).

*How many instances are there in total (of each type, if appropriate)?*

We provide data from 40 teams.

*Does the dataset contain all possible instances or is it a sample (not necessarily random) of instances from a larger set?* If the dataset is a sample, then what is the larger set? Is the sample representative of the larger set (e.g., geographic coverage)? If so, please describe how this representativeness was validated/verified. If it is not representative of the larger set, please describe why not (e.g., to cover a more diverse range of instances, because instances were withheld or unavailable).

The dataset is a sample of the subset of the human population that meet the following criteria: (i) over 18 years of age, (ii) able to read and speak English, and does not have any major physical limitations that would interfere with completing tasks on a computer (e.g., limited vision or hearing, problems with fine motor control).

The sample is weighted heavily towards younger participants (ages 18–22, corresponding to the typical age range for undergraduates at the university where the study was conducted). The ratio of

female to male participants is representative of the sex distribution of the students at the university [123].

Of the participants whose race was recorded, 48% identified as 'Non-Hispanic White' or 'European American', which is lower than the proportion of enrolled students at the university that identified as 'White' (66%). 7% identified as 'African American' and 13% as 'Asian American', which is roughly consistent with the proportion of students at the university who identify as 'Black or African American' (6.7%) or 'Asian' (10.6%) of the student population at the university. 27% of the participants identified as Hispanic, which is consistent with the portion of students at the university who identify as 'Hispanic or Latinx', and greater than the percentage of 'Hispanic or Latino' people in the US population in the 2020 census (18.7%) [124]—this can be attributed to the geographic location of the university (the southwestern US). Most deviations from representativeness of the sample from the larger population can be attributed primarily to the sampling methods we used (convenience and snowball sampling, described later in this document).

*What data does each instance consist of?* "Raw" data (e.g., unprocessed text or images) or features? In either case, please provide a description.

Modalities include two kinds of brain scan data—functional near-infrared spectroscopy (fNIRS) and electroencephalography (EEG), as well as skin conductance, heart rate, eye tracking, face images, spoken dialog audio data with automatic speech recognition (ASR) transcriptions, game screenshots, gameplay data, game performance data, demographic data, and self-report questionnaires.

We provide both nearly-raw and derived data products.

*Is there a label or target associated with each instance?* If so, please provide a description.

There are many potential prediction tasks that can be performed with this dataset. Some that may be of interest to researchers are provided below:

- Individual and team scores on the baseline tasks.
- Post-game survey questionnaire responses.

*Is any information missing from individual instances?* If so, please provide a description, explaining why this information is missing (e.g., because it was unavailable). This does not include intentionally removed information, but might include, e.g., redacted text.

There are instances of missing data in the dataset. Some of the instances are expected, as it is built into the design of the experiment (e.g., when participants skip optional questions in questionnaires). However, as is the case with any complex study involving human subjects, there are many ways for data collection to not proceed as originally expected, resulting in missing data. We have instances of missing data due to a variety of reasons:

- When an experimenter sat in for a no-show participant or a participant that left in the middle of a group session, most of the time, the experimenter did not wear an EEG/fNIRS cap or eye tracker. This is because properly setting up the cap and calibrating the eye tracker for a participant is a time-consuming process, and so in such cases, we chose to prioritize the number of tasks for which we had data for the group session over ensuring the presence of the EEG/fNIRS/gaze modalities. Thus, we do not obtain physiological measures for them for the whole group session.
- There were a few group sessions where a participant either refused to wear their EEG/fNIRS cap (and/or eye tracker) or had to take them off due to physical discomfort, usually caused by the cap not setup right for the participant's head size, or glasses not fitting right on their head or over their own eyeglasses. In these cases, the physiological measures were not obtained for the entire group session for that participant.
- Participant takes cap or glasses off sometime in the middle of the experiment after tasks have been completed: Again, this was usually caused by the participant not feeling comfortable with the device on, the cap not setup right for the participant's head size, glasses not fitting right on head or over personal glasses, or the participant feeling sick. Therefore, the EEG,

NIRS, and/or gaze data would be good for completed tasks up to the point where the participant took the device off.

- The following group sessions had no EEG amplifier at the 'lion' station: exp_2023_04_17_13, exp_2023_04_18_14, exp_2023_04_20_14, exp_2023_04_21_10, exp_2023_04_24_13, exp_2023_04_26_10, and exp_2023_04_27_14. We had to send the amplifier in for repair around 2023-04-17 and did not receive a loaner amplifier until 2023-04-28. Therefore, the EEG data for the 'lion' station for those group sessions is either missing or invalid.

- In earlier group sessions, we attempted to run the NIRx Aurora fNIRS data acquisition software on an iMac. However, it would crash sometimes. If we noticed the crash, in most cases, we would immediately go to the iMac and restart it. However, there were some sessions where we did not find out that Aurora had crashed until after the session had ended, leading to missing fNIRS data for these sessions. In later group sessions, we switched to using a Windows computer, which resolved the problem.

*Are relationships between individual instances made explicit (e.g., users' movie ratings, social network links)?* If so, please describe how these relationships are made explicit.

Each participant has a unique ID, as does each group session. These can be used to link data from different modalities and tasks.

*Are there recommended data splits (e.g., training, development/validation, testing)?* If so, please provide a description of these splits, explaining the rationale behind them.

We do not provide recommended data splits.

*Are there any errors, sources of noise, or redundancies in the dataset?* If so, please provide a description.

**Sources of noise** EEG signals are susceptible to artifacts from eye movements/blinks [125] and external sources like fluorescent lights or grounding problems [126]. From plotting the raw EEG data, we observed a predictable 60 Hz electrical noise in the signal (and related harmonics). We also observed a peak around 5 Hz that may be due to a grounding issue or some other environmental influence. Using MNE-Python [127] we eliminated noise from these sources using a notch filter with frequency set to 60 Hz, trans bandwidth set to 9 Hz and notch widths set to 2 Hz.

fNIRS signals are known to have motion artifacts (MA) (e.g., cardiac and respiratory artifacts). MA of all fNIRS channels' oxyhemoglobin (HbO) and deoxyhemoglobin (HbR) concentrations were then filtered using a bandpass filter with bandwith set to 0.01–0.2Hz [128].

EKG data may contain high-frequency noise and artifacts due to muscle movements. These were filtered using a filter with highpass: 0.0 Hz lowpass: 250.0 Hz using MNE-Python [127].

The GSR data may contain high-frequency noise and rapid-transient artifacts, these were filtered using a filter with highpass: 0.0 Hz lowpass: 250.0 Hz using MNE-Python [127].

*Is the dataset self-contained, or does it link to or otherwise rely on external resources (e.g., websites, tweets, other datasets)?* If it links to or relies on external resources, a) are there guarantees that they will exist, and remain constant, over time; b) are there official archival versions of the complete dataset (i.e., including the external resources as they existed at the time the dataset was created); c) are there any restrictions (e.g., licenses, fees) associated with any of the external resources that might apply to a dataset consumer? Please provide descriptions of all external resources and any restrictions associated with them, as well as links or other access points, as appropriate.

The dataset is self-contained. However, documentation for the schemas of the messages that comprise the data from the testbed is not on our website but is available in the repository for the ASIST Study 3 testbed [4].

*Does the dataset contain data that might be considered confidential (e.g., data that is protected by legal privilege or by doctor-patient confidentiality, data that includes the content of individuals' non-public communications)?* If so, please provide a description.

The dataset does not contain data that might be considered confidential.

*Does the dataset contain data that, if viewed directly, might be offensive, insulting, threatening, or might otherwise cause anxiety?* If so, please describe why.

The data contains screenshots of a Minecraft-based urban search-and-rescue mission involving rescuing victims that were injured in a building collapse. While we expect that these are unlikely to cause anxiety due to the low visual fidelity of Minecraft, we are not able to rule it out.

*Does the dataset identify any subpopulations (e.g., by age, gender)?* If so, please describe how these subpopulations are identified and provide a description of their respective distributions within the dataset. Is it possible to identify individuals (i.e., one or more natural persons), either directly or indirectly (i.e., in combination with other data) from the dataset? If so, please describe how.

The dataset includes questions on demographics, gaming experience, and health. Depending on the variables on interest, it is possible for researchers to look at data by gender, race/ethnicity, and age, among other variables. However, no one was purposefully excluded based on any of these criteria.

To the best of our knowledge, it is not possible to identify individuals either directly or indirectly from the publicly released portions of this dataset.

Aggregated demographic data for selected dimensions are provided in Table 3. Note that demographic data is missing for two regular participants and the 9 experimental confederates that filled in for missing participants.

Table 3: Aggregated demographic data for selected dimensions.

(a) Age distribution

(b) Sex

| | |
|---|---|
| Female | 54 |
| Male | 45 |
| Prefer not to say | 1 |
| Other | 0 |

(c) Race

| | |
|---|---|
| Other | 32 |
| Non-Hispanic White | 28 |
| European American | 20 |
| Asian American | 13 |
| African American | 7 |

(d) Hispanic

| | |
|---|---|
| No | 73 |
| Yes | 27 |

(e) Videogaming experience

| | |
|---|---|
| Have played them occasionally | 38 |
| Have played them fairly often | 30 |
| Have played them regularly for years | 28 |
| Never played them | 4 |

(f) Minecraft experience

| | |
|---|---|
| Have played it occasionally | 42 |
| Have played it fairly often | 24 |
| Have played it regularly for years | 18 |
| Never played it | 16 |

*Does the dataset contain data that might be considered sensitive in any way (e.g., data that reveals race or ethnic origins, sexual orientations, religious beliefs, political opinions or union memberships, or locations; financial or health data; biometric or genetic data; forms of government identification, such as social security numbers; criminal history)?* If so, please provide a description.

All publicly released data in the dataset has been deidentified.

**Questionnaires** While we collected socio-demographic and health data through self-report, any identifiable information has been removed so the data we provide does not include personal information (e.g., e-mail, phone number, etc.) for any of the participants.

**Audio**    We recorded spoken audio data from participants during the study. While participants have given their consent for their audio recordings to be shared publicly, we do not do so at this time since we believe that the audio data is still fairly sensitive. We will publish derived features from the audio data (e.g., vocalic features extracted using openSMILE [129]). We will support researchers who want to perform alternative processing on the raw audio files by working with them to run their processing code on our machines and sharing the resulting derived data with them.

**Face images**    We do not plan to release the face images that we captured during the experimental sessions, as they contain personally identifiable information (PII). We plan to provide automated FACS codes for the images by processing them using OpenFace [130], contingent on the codes passing a manual validation/spot-checking procedure[1]. Similar to the raw audio, we are willing to work with researchers to run their custom face image processing pipelines on our machines and share the (de-identified) results with them.

*Any other comments?*

None.

### A.3    Collection process

*How was the data associated with each instance acquired?* Was the data directly observable (e.g., raw text, movie ratings), reported by subjects (e.g., survey responses), or indirectly inferred/derived from other data (e.g., part-of-speech tags, model-based guesses for age or language)? If the data was reported by subjects or indirectly inferred/derived from other data, was the data validated/verified? If so, please describe how.

The data was directly observable in some case (e.g., raw audio/images/physiological measures), and reported by subjects for other cases (e.g. self-report questionnaires). For the data that was reported by subjects, we used questionnaires that were previously validated in the literature.

*What mechanisms or procedures were used to collect the data (e.g., hardware apparatuses or sensors, manual human curation, software programs, software APIs)?* How were these mechanisms or procedures validated?

See Appendix H for details on how the data was collected.

*If the dataset is a sample from a larger set, what was the sampling strategy (e.g., deterministic, probabilistic with specific sampling probabilities)?*

The sampling strategy for this study was a convenience sample, which is a non-probabilistic sampling strategy where participants are included due to ease of access. We also used snowball sampling as participants were asked to pass along our study contact information to any of their friends, peers, co-workers, and other people who they thought might be interested in participating.

*Who was involved in the data collection process (e.g., students, crowdworkers, contractors) and how were they compensated (e.g., how much were crowdworkers paid)?*

An interdisciplinary research team comprised of individuals from Computer Science (CS), Human Development & Family Science (HDFS), Psychology, and Speech, Language, and & Hearing Sciences (SLHS) was involved in the data collection process. A postdoctoral research associate in HDFS recruited and scheduled participants for the study, among other study-related responsibilities. Data collection was conducted by a team of graduate students who working hands-on and oversaw undergraduate research assistants (RAs). All graduate students were financially compensated as this project provided funding in the form of a research assistantship. However, undergraduate students were compensated in the form of independent research study credit with the study PI. Undergraduates

---

[1]Participants wore masks throughout the experiment, which poses a significant challenge for automated FACS coding.

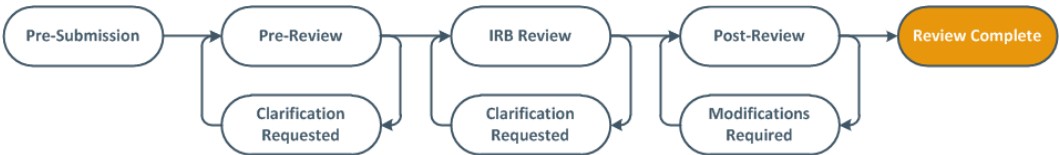

Figure 2: Workflow showing the IRB process for human subjects research.

.

enrolled in the independent study as part of their semester coursework and received 3 credits at the end of the semester in a graded format (pass/fail).

Participants were compensated in the form of an electronic Amazon.com gift card. Compensation was calculated based on time spent in the study—$5.00 for the 30-minute pre-session and $10.00 for each hour of the testing session for a maximum of $35.00.

Participants were further incentivized through gamification—they had a chance of winning an additional $20.00 Amazon gift card if they obtained the highest score on the Minecraft missions out of 20 group sessions.

Participants who enrolled in the study through the SONA system received compensation in the form of SONA credit. For the pre-session and testing session, participants received 1 credit and 3 credits, respectively.

*Over what timeframe was the data collected?* Does this timeframe match the creation timeframe of the data associated with the instances (e.g., recent crawl of old news articles)? If not, please describe the timeframe in which the data associated with the instances was created.

The data was collected between September 9, 2022, and March 5, 2023.

*Were any ethical review processes conducted (e.g., by an institutional review board)?* If so, please provide a description of these review processes, including the outcomes, as well as a link or other access point to any supporting documentation.

Yes, the project (IRB 2001272896) was approved by the institution (see approval below).

Prior to beginning the study, an in-depth ethical review process was conducted by the University of Arizona's Institutional Review Board (IRB). This step requires the study PI(s) to complete the application for human subjects research and provide any documentation that will be used during the study including, but not limited to, the following:

- List of all members of the research team
- Updated Collaborative Institutional Training Initiative (CITI) and Conflict of Interest (COI) trainings for all members of the research team.
- Informed consent forms
- E-mails to participants
- Advertisements
- Phone call/e-mail script
- Debriefing
- Study questionnaires
- Research design and methods

The human subjects research application is submitted to our electronic IRB system (eIRB) where it undergoes the workflow shown in Figure 2.

The IRB reviewer and the study PI and/or primary contact for IRB on the research team's side are always in communication to address requested edits. When all edits have been completed, the project goes to the full IRB for review. At this stage, more clarifications may be requested or the project goes to post-review. Once everything is finalized and the IRB approval is on all documents, then those

documents are used until they are updated (e.g., when modifications are necessary, continuing review, etc.).

The recruitment materials submitted to the IRB are included in this document (see Appendix P).

### *Did you collect the data from the individuals in question directly, or obtain it via third parties or other sources (e.g., websites)?*

We collected the data from the individuals in question directly.

### *Were the individuals in question notified about the data collection?* If so, please describe (or show with screenshots or other information) how notice was provided, and provide a link or other access point to or otherwise reproduce, the exact language of the notification itself.

The informed consent form explained to participants the data collection process in depth. In addition, the research team would always ensure that the participants were aware of the various parts of data collection (e.g., fNIRS, EEG, REDCap survey, etc.) and had opportunity to ask questions at any point in time. At the beginning and throughout each session, the research team went over what will be asked of the participants. The research team were always communicating with the participants and vice versa about what data we were going to collect in a specific part of the study.

Participants also were able to contact the research team whenever they had questions prior to coming in for the in-lab sessions. The questions would range from ensuring they understood the duration of the pre-session and testing session to questions about the consent form to scheduling needs and verifying study location.

### *Did the individuals in question consent to the collection and use of their data?* If so, please describe (or show with screenshots or other information) how consent was requested and provided, and provide a link or other access point to, or otherwise reproduce, the exact language to which the individuals consented.

Informed consent was obtained in-person when participants visited the University for their initial lab session (referred to as the pre-session). When participants arrived at the research lab a trained undergraduate RA or graduate student went over the consent form which covered all study-related information. Then the RA asked the participant to read/look over the consent form to ensure that the participant understood the study and was able to ask any questions or concerns. The RA reminded the participant that they are not required to participate and can refuse to consent without penalty.

We also conducted informed consent online through REDCap. Participants who were interested in the study and agreed to participate were sent an individualized link for the consent form in REDCap. The consent form was housed in the File Repository in REDCap which is secure. When participants came for the in-lab sessions, they were again asked if they had any questions or concerns about the study. If participants had not completed the consent form prior to their in-lab sessions, then their individualized link was opened on the computer and they completed the consenting process then. The experimenter conducting the consent reminded the participants that they were not required to participate and could refuse to consent without penalty.

For any in-person informed consent, the research team left the participant alone in the room so that the participant could carefully read through the consent form and would not feel pressured to respond due to timing, presence of others, etc.

The consent forms are included in this document (see Appendix O).

### *If consent was obtained, were the consenting individuals provided with a mechanism to revoke their consent in the future or for certain uses?* If so, please provide a description, as well as a link or other access point to the mechanism (if appropriate).

Participants were notified throughout the consent and during the in-lab sessions by the research team that they could opt out of the study at any point in time. In such circumstances, the participants had access to the research team through e-mail, phone, and in-person so that they could withdraw from the study.

Consent for future research was requested on the informed consent form for the present UA research team to store the identifiable data indefinitely (video and audio recording; eye tracking data) and to use it for future research without obtaining additional consent.

We recently had to change the PI of record on the IRB protocol since the previous PI of record unexpectedly passed away. We are preparing to submit a modification to contact all study participants and notify them of the contact information for the new PI. Once this has been approved, we will be updating the participants with this contact information.

***Has an analysis of the potential impact of the dataset and its use on data subjects (e.g., a data protection impact analysis) been conducted?*** If so, please provide a description of this analysis, including the outcomes, as well as a link or other access point to any supporting documentation.

While we did not perform a formal data protection impact analysis, the IRB approval process required us to describe impacts on study participants. The UArizona IRB deemed the risks to participants from our study to be minimal.

### A.4  Preprocessing/cleaning/labeling

***Was any preprocessing/cleaning/labeling of the data done (e.g., discretization or bucketing, tokenization, part-of-speech tagging, SIFT feature extraction, removal of instances, processing of missing values)?*** If so, please provide a description. If not, you may skip the remaining questions in this section.

**EEG electrode setup alteration**   In our study, we made an alteration to our EEG electrode setup for experiments on or after November 11, 2022. Prior to this date, our EEG data comprised several channels, including AFF5h, FC1, CP5, CP1, PO9, Oz, PO10, CP6, CP2, FC2, and AFF6h. However, due to the prolonged calibration time required for these electrodes, we opted to exclude them from our experimental sessions starting November 11, 2022. The montage in Figure 5 reflects the updated setup.

For experiments prior to November 11, 2022, our dataset contains the following EEG channels: AFF1h, F7, FC5, C3, T7, TP9, Pz, P3, P7, O1, O2, P8, P4, TP10, Cz, C4, T8, FC6, FCz, F8, AFF2h, AFF5h, FC1, CP5, CP1, PO9, Oz, PO10, CP6,CP2, FC2, AFF6h. For experiments on or after November 11, 2022, our dataset contains the following EEG channels: AFF1h, F7, FC5, C3, T7, TP9, Pz, P3, P7, O1, O2, P8, P4, TP10, Cz, C4, T8, FC6, FCz, F8, AFF2h (omitting AFF5h, FC1, CP5, CP1, PO9, Oz, PO10, CP6,CP2, FC2, AFF6h) to meet the time constraint of each experiment.

Note that while the raw EEG data collected before November 11, 2022 contains a greater number of channels, for the purposes of consistency and comparative analysis, we confined our analysis to the channels that were present in the data both before and after the mentioned date. The channels we retained and used in the valence/arousal prediction benchmark analysis are the following: AFF1h, F7, FC5, C3, T7, TP9, Pz, P3, P7, O1, O2, P8, P4, TP10, Cz, C4, T8, FC6, FCz, F8, AFF2h. By streamlining our channels, we aimed to ensure more efficient data collection and analysis, while maintaining the rigor and validity of our findings.

The following preprocessing and cleaning steps were performed for the data in the SQLite database:

**Minecraft issues**   For instances where a Minecraft mission has to be restarted due to technical issues, we remove the data from the previous (interrupted) mission, and only keep the data from the new (uninterrupted) mission.

**Physiological measure data for confederates**   We remove physiological measure data from group session tasks in which a confederate stepped in to fill the role of a no-show participant or a participant that left in the midst of a group session if that confederate did not put on an EEG/fNIRS cap or eye tracker.

***Was the "raw" data saved in addition to the preprocessed/cleaned/labeled data (e.g., to support unanticipated future uses)?*** If so, please provide a link or other access point to the "raw" data.

The raw data is saved on our lab servers. While we do not rule out sharing it (sans the identifiable data) with the public in the future, we believe that its size (just shy of 20 TB), complexity, and unclean nature makes its utility to downstream consumers limited enough to not justify the considerable logistical effort required to share it at the moment.

We do provide cleaned data though through our SQLite database, which consists of of non-PII data that has been checked to be valid. We also provide the scripts that were used to clean the data.

***Is the software that was used to preprocess/clean/label the data available?*** If so, please provide a link or other access point.

The software used to preprocess/clean/label the data is available at `https://github.com/ml4ai/tomcat`.

- The code for building the SQLite database from the raw data is in the `human_experiments/datasette_interface` directory.
- The code for creating the processed datasets for the benchmark analyses are in the `human_experiments/lab_software/tomcat-physio-data-extraction` and `tomcat-physio-data-extraction-v2` directories. The results from these are then further processed by the code in the `data_products/scripts` directory. This code effectively segmented the data into manageable chunks, specifically isolating the task data relevant to our benchmark analyses. Also, the code for constructing the synchronized physio data from the raw data is in the `human_experiments/lab_software/tomcat-physio-synch` directory. This synchronizes all the physio data.

***Any other comments?***

None.

### A.5 Uses

***Has the dataset been used for any tasks already?*** If so, please provide a description.

The dataset has not been used for any tasks already (besides the benchmark analyses in the main paper).

***Is there a repository that links to any or all papers or systems that use the dataset?*** If so, please provide a link or other access point.

Links to papers and systems that use the dataset are listed at `https://tomcat.ivilab.org/papers-and-systems`.

***What (other) tasks could the dataset be used for?***

With some additional annotations effort, the dataset could also be used for the following tasks:

- Sentiment analysis
- Emotion recognition
- Plan recognition
- Entrainment detection

***Is there anything about the composition of the dataset or the way it was collected and preprocessed/cleaned/labeled that might impact future uses?*** For example, is there anything that a dataset consumer might need to know to avoid uses that could result in unfair treatment of individuals or groups (e.g., stereotyping, quality of service issues) or other risks or harms (e.g., legal risks, financial harms)? If so, please provide a description. Is there anything a dataset consumer could do to mitigate these risks or harms?

Future analyses of the derived data products may be affected by the specific filtering/interpolation strategies used. We do not currently know of specific uses of the dataset that could result in unfair treatment of individuals or groups.

*Are there tasks for which the dataset should not be used?* If so, please provide a description.

We do not currently know of specific tasks for which the dataset should not be used, but users should in general strive to make sure that their usage of the dataset does not perpetuate systemic inequalities. The dataset should not be used to attempt to re-identify individual participants.

*Any other comments?*

None.

## A.6 Distribution

*Will the dataset be distributed to third parties outside of the en- tity (e.g., company, institution, organization) on behalf of which the dataset was created?* If so, please provide a description.

The dataset will be distributed publicly.

*How will the dataset will be distributed (e.g., tarball on website, API, GitHub)?* Does the dataset have a digital object identifier (DOI)?

The dataset is available at `https://tomcat.ivilab.org`. See Appendix C for more details.

We have not set up a DOI for the dataset yet, since the size of the dataset is too large for commonly used DOI providers (e.g., Zenodo, Figshare). We are working with CyVerse to set up a DOI. In any case, we do not expect the current URL (`https://tomcat.ivilab.org`) to change in the foreseeable future.

*When will the dataset be distributed?*

The initial release of the dataset is already public at `https://tomcat.ivilab.org`. We will add additional raw and derived data in the near future.

*Will the dataset be distributed under a copyright or other intel- lectual property (IP) license, and/or under applicable terms of use (ToU)?* If so, please describe this license and/or ToU, and provide a link or other access point to, or otherwise reproduce, any relevant licensing terms or ToU, as well as any fees associated with these restrictions.

The data is licensed under the Creative Commons Attribution-NonCommercial-ShareAlike 4.0 International (CC BY-NC-SA 4.0) license (`https://creativecommons.org/licenses/by-nc-sa/4.0`). The software is licensed under the MIT License.

*Have any third parties imposed IP-based or other restrictions on the data associated with the instances?* If so, please describe these restrictions, and provide a link or other access point to, or otherwise reproduce, any relevant licensing terms, as well as any fees associated with these restrictions.

No third parties have imposed restrictions on the data associated with the instances.

*Do any export controls or other regulatory restrictions apply to the dataset or to individual instances?* If so, please describe these restrictions, and provide a link or other access point to, or otherwise reproduce, any supporting documentation.

The dataset is not subject to export control or other regulatory restrictions.

*Any other comments?*

None.

## A.7 Maintenance

### *Who will be supporting/hosting/maintaining the dataset?*

The responsibility for supporting the dataset will lie with the ToMCAT project principle investigators Adarsh Pyarelal, Clayton Morrison, and Kobus Barnard. The data will be initially hosted on a web server in the Computer Science department at U. Arizona, and will be mirrored by CyVerse as curated data. CyVerse (`https://cyverse.org`) is a cyber infrastructure system led by U. Arizona, initiated by NSF in 2008, and has external and institutional support for curating data indefinitely.

### *How can the owner/curator/manager of the dataset be contacted (e.g., email address)?*

The maintainers can be contacted by email: `tomcat-dataset-maintainers@list.arizona.edu`.

### *Is there an erratum?* If so, please provide a link or other access point.

Errata will be published at `https://tomcat.ivilab.org/errata`.

### *Will the dataset be updated (e.g., to correct labeling errors, add new instances, delete instances)?* If so, please describe how often, by whom, and how updates will be communicated to dataset consumers (e.g., mailing list, GitHub)?

The dataset will be updated periodically by the maintainers, with three types of updates.

| Type of update | Frequency |
|---|---|
| Adding new raw data (e.g., from additional modalities as they get processed) | We expect a number of updates to be published in summer and fall 2023 as we process more of the raw data and upload data from additional modalities. It is unlikely (but not impossible) that we will add data from new participants to this particular dataset, post fall 2023. |
| Fixing previously uploaded data. | This is likely to occur often in summer and early fall 2023, and very infrequently post 2023. |
| Adding new derived data (e.g., from improved interpolation procedures) | We expect this to happen relatively infrequently, triggered by either (i) a new publication on the data or (ii) a request from a collaborator for a particular form of derived data. |

Updates will be communicated to dataset consumers via the mailing list `tomcat-dataset-updates@list.arizona.edu`. They will also be published at `https://tomcat.ivilab.org/updates`.

### *If the dataset relates to people, are there applicable limits on the retention of the data associated with the instances (e.g., were the individuals in question told that their data would be retained for a fixed period of time and then deleted)?* If so, please describe these limits and explain how they will be enforced.

There are no limits on the retention of the data.

### *Will older versions of the dataset continue to be supported/hosted/maintained?* If so, please describe how. If not, please describe how its obsolescence will be communicated to dataset consumers.

We will rename and/or move older versions of the data to an archive location. As appropriate, we will communicate such changes to dataset consumers via the mailing list `tomcat-dataset-updates@list.arizona.edu` and at `https://tomcat.ivilab.org/updates`. We will continue to host older versions for a minimum of two years, and we will only remove data if it is judged to have no value moving forward. Obsolescence of such versions will be communicated to dataset consumers as above.

Others wishing to extend/augment/build on/contribute to the dataset will be encouraged to email us at `tomcat-dataset-maintainers@list.arizona.edu` about their plans.

We will validate proposed contributions to some extent before integrating them into the hosted data. One possible category of contribution is a new way distill, transform, or represent the raw data, leading to a new derived data tables. Here we will require the code to replicate producing the new data. Another possible contribution would be a labeling of the data, such as manual transcription of speech data, or human evaluation of emotive speech. Here we will compare with existing versions and/or perform spot checks.

None.

## B    Recruitment

Participants were recruited at the University of Arizona. All advertisement and recruitment materials provided contact information for the Temporal Interpersonal Emotion Systems (TIES) Lab and the corresponding Gmail address and Google Voice phone number. IRB-approved flyers were posted across campus and shared on social media and the project's website (see Appendix P).

To further reach the undergraduate student body at the university, study information was placed on the SONA system for Psychology as the project's PIs were affiliate faculty in the department. This allowed students to either receive credit for research experience in their lower-level Psychology courses and/or receive extra credit for participation in research experience in their Family Studies & Human Development (FSHD) courses.

Lastly, we also recruited through word-of-mouth and sending e-mails with IRB-approved language to undergraduate, graduate student, and faculty listservs and/or newsletters.

## C    Data format

The dataset uses two open and widely used data formats—namely, SQLite and CSV, both of which are recommended storage formats for datasets according to the Library of Congress [131].

We provide access to the data in two ways. The first is through a Datasette [121] instance, which provides graphical and programmatic interfaces for users to perform SQL queries to retrieve subsets of data they are interested in, or simply download the backing SQLite database. The second is through links to pre-built files containing selected parts of the data that we (i) used for the experiments in this paper, and (ii) expect will be commonly requested as good starting points for analysis. We provide all valid raw data and several derived data products. One key issue with using the raw data is that it is it is comprised of data from multiple asynchronous data streams. Hence, aligning the data for multiple participants entails interpolation, which we typically do for most of the derived data sets. A second issue is recording specific issues encountered during data collection, such as an experimenter stepping in for a participant who decided to leave in the midst of a group session. Finally, derived data will typically also include standard data transformations and cleaning. More details are available on the website itself.

## D    Structured metadata

Structured metadata using web standards (`schema.org`) has been added to the index page for `https://tomcat.ivilab.org`.

We have also added a route to the Datasette instance to enable quick inspection and programmatic access to the structured metadata: `https://tomcat.ivilab.org/-/structured_metadata.json`.

# E  Long-term preservation

We are committed to long term preservation and availability of this data. We have redundancy in our maintainers, all three of whom are faculty at the University of Arizona. We are able to host the data on new servers in two different academic units. Initially, we will host the primary data source on a web server maintained by the U. Arizona Computer Science Department. We will also mirror the data on the CyVerse curated data repository. CyVerse (`https://cyverse.org`) is a cyber infrastructure system led by U. Arizona, initiated by NSF in 2008, and has external and institutional support for curating data indefinitely.

# F  Reproducibility

We use the ML reproducibility checklist v2.0 (`https://www.cs.mcgill.ca/~jpineau/ReproducibilityChecklist.pdf`). Since we do not present any new models, algorithms, or theoretical claims in this work, we skip the corresponding items in the checklist.

## F.1  CNN experiments

fNIRS and EEG were trained and tested separately using Convolutional Neural Networks (CNN) models.

For all datasets used, check if you include:

- ✓ **Relevant statistics** We use the data from the individual affective task, encompassing five distinct classes (-2, -1, 0, 1, 2) that represent valence and arousal scores. The dataset consists of fNIRS recordings obtained from 102 subjects and EEG recordings obtained from 97 subjects. The disparity in subject counts can be attributed to instances where the EEG amplifier encountered technical issues, resulting in a failure to record data for certain individuals.

- ✓ **Details regarding train/validation/test splits.** To ensure robust model evaluation, the dataset is divided into an 80% training set and a 20% testing set, employing a 5-fold cross-validation approach, where each fold acts as a validation set for training the model.

- ✓ **An explanation of any data that were excluded, and all pre-processing steps.** Within the affective task dataset, encompassing both EEG and fNIRS data, we find two essential columns: `valence_score` and `arousal_score`. These columns capture the participant's subjective ratings of images during specific time intervals. However, it is important to note that numerous entries within these columns may currently lack values and remain empty.

  We used the `pandas` [132, 133] `fillna` function with the `bfill` method in this code to fill those empty entries with appropriate values. By using `bfill`, the missing values in these columns were filled with the next available value from the same column, going backwards.

  This ensures that the missing values are replaced with values that are likely to be similar to the previous ratings given by the participants.

  To transform signals into images for CNN processing, we may need to perform an offset, particularly when accounting for the Hemodynamic Response Factor in fNIRS data. This offset determines the starting point for data extraction corresponding to each image viewed by the participant. For example, setting an offset of 10 means that the first 10 rows of data for each unique stimulus-response instance are skipped, and extraction begins from the 11th row. By default, the offset is set to 0.

  To create images, we must consider multiple rows of data, defined by the window size parameter. Setting a window size results in the extraction of a data segment for the length of the window size following the stimulus presentation. For example, if the window size is set to 10, it would extract 10 rows of data from the segment in which the participant is rating an image.

  The images generated are subsequently annotated based on the predominant valence and arousal scores observed within their respective windows.

- ✓ **A link to a downloadable version of the dataset or simulation environment.** The dataset used for these experiments is available at `https://tomcat.ivilab.org/`

Table 4: CNN architecture for fNIRS data. $W$ is the size of the window.

| Layer (Type) | Output shape | # of parameters |
|---|---|---|
| Conv2d-1 | [-1, 16, $W$, 22] | 160 |
| BatchNorm2d-2 | [-1, 16, $W$, 22] | 32 |
| MaxPool2d-3 | [-1, 16, $\frac{W}{2}$, 11] | 0 |
| Conv2d-4 | [-1, 32, $\frac{W}{2}$, 11] | 12,832 |
| BatchNorm2d-5 | [-1, 32, $\frac{W}{2}$, 11] | 64 |
| MaxPool2d-6 | [-1, 32, $\frac{W}{4}$, $\frac{W}{4}$] | 0 |
| Conv2d-7 | [-1, 64, $\frac{W}{4}$, $\frac{W}{4}$] | 100,416 |
| BatchNorm2d-8 | [-1, 64, $\frac{W}{4}$, $\frac{W}{4}$] | 128 |
| MaxPool2d-9 | [-1, 64, $\frac{W}{8}$, $\frac{W}{8}$] | 0 |
| AdaptiveAvgPool2d-10 | [-1, 64, 1, 1] | 0 |
| Linear-11 | [-1, 128] | 8,320 |
| Dropout-12 | [-1, 128] | 0 |
| Linear-13 | [-1, 64] | 8,256 |
| Linear-14 | [-1, 5] | 325 |
| Linear-15 | [-1, 5] | 325 |

`derived-data-products`. After extracting the data, you will find two folders named `fnirs_10hz` and `eeg_500hz`. Within each of these folders, there will be an experiment folder. Inside the experiment folder, you will find files named `affective_individual_*.csv`.

- ✓ **For new data collected, a complete description of the data collection process, such as instructions to annotators and methods for quality control.** The data collection process is detailed in the main paper and in Appendix H.

For all shared code related to this work, check if you include:

- ✓ **Specification of dependencies.** The dependencies are listed in the `requirements.txt` file of the directory `code/ToMCAT-ML-Modeling` in the supplementary material.

- ✓ **Training code.** The training code is included with the supplementary materials (see the directory `code/ToMCAT-ML-Modeling`)

- ✓ **Evaluation code.** The evaluation code is included with the supplementary materials (see the directory `code/ToMCAT-ML-Modeling`).

- ✓ **(Pre-)trained model(s).** We do not provide pre-trained models, since our CNN models are relatively small and quick to train, and since we provide the training data as well.

- ✓ **README file includes table of results accompanied by a precise command to run to produce those results.**

For all reported experimental results, check if you include:

- ✓ **The range of hyper-parameters considered, method to select the best hyper-parameter configuration, and specification of all hyper-parameters used to generate results.**
  The architecture and configuration of the CNN models we used are summarized in Table 4 and Table 5. Notably, the EEG CNN model incorporates an input shape of [W,56]. Here, $W$ represents the window size. The value 56 is derived from 7 EEG channels, each divided into 4 frequency bands (Theta, Alpha, Beta, Gamma). Thus, we obtain 28 values (4 frequency bands per channel). From these, wavelet features are extracted and then horizontally appended, resulting in 56 columns of EEG data. In contrast, the fNIRS CNN model has an input size of [W,22], which represents the 11 HbO and 11 HbR fNIRS channels considered. Both the EEG and fNIRS CNN models shared a common number of classes, set to 5.

  For the fNIRS CNN model with a 2-second window and a 2-second offset, the optimal configuration was found with 15 epochs and a batch size of 50. This configuration, specifically for the 0-second offset, achieved an accuracy of 30.4 ± 1.2 for valence and 30.3 ± 0.8 for arousal, with a loss of 3.02 ± 0.008. The confusion matrices for fNIRS arousal and valence are presented in Figure 3.

Table 5: CNN architecture for EEG data. $W$ is the size of the window.

| Layer (Type) | Output shape | # of parameters |
|---|---|---|
| Conv2d-1 | [-1, 16, W, 56] | 2,000 |
| BatchNorm2d-2 | [-1, 16, W, 56] | 32 |
| Conv2d-3 | [-1, 32, W-27, 56] | 896 |
| BatchNorm2d-4 | [-1, 32, W-27, 56] | 64 |
| ELU-5 | [-1, 32, W-27, 56] | 0 |
| AvgPool2d-6 | [-1, 32, W-27, 14] | 0 |
| Dropout-7 | [-1, 32, W-27, 14] | 0 |
| Conv2d-8 | [-1, 32, W-27, 15] | 16,384 |
| BatchNorm2d-9 | [-1, 32, W-27, 15] | 64 |
| ELU-10 | [-1, 32, W-27, 15] | 0 |
| AvgPool2d-11 | [-1, 32, W-27, 3] | 0 |
| Dropout-12 | [-1, 32, W-27, 3] | 0 |
| Linear-13 | [-1, 128] | 11,956,352 |
| ELU-14 | [-1, 128] | 0 |
| Linear-15 | [-1, 5] | 645 |
| Linear-16 | [-1, 5] | 645 |

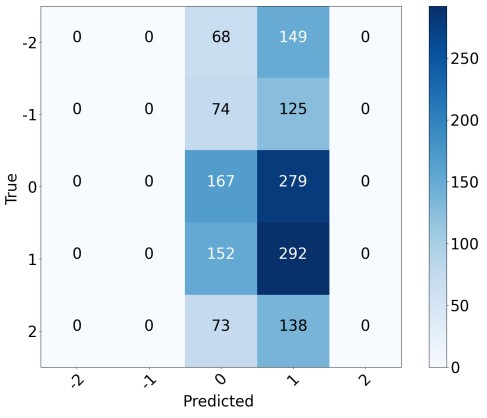

(a) fNIRS Arousal Confusion Matrix

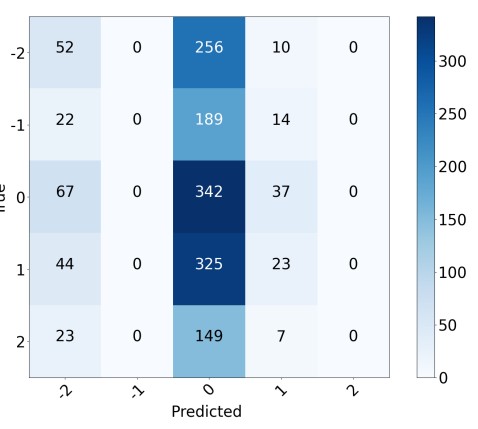

(b) fNIRS Valence Confusion Matrix

Figure 3: Confusion matrices for arousal and valence prediction based on fNIRS data with a 1-second window and a 2-second offset.

Regarding EEG CNN model with a 1-second window and a 0-second offset, the optimal configuration was found with 15 epochs and a batch size of 50. This configuration, specifically achieved an accuracy of 29.3 ± 1.4 for valence and 29.8 ± 1.3 for arousal, with a loss of 2.8 ± 0.004. The confusion matrices for EEG arousal and valence are shown in Figure 4.

The hyperparameters used for the fNIRS and EEG CNN models include a learning rate of 0.01, a batch size of 50, and training for 15 epochs.

✓ **The exact number of training and evaluation runs.** We conducted a total of 7 training and evaluation runs for each model. These runs were performed using a 5-fold cross-validation strategy, ensuring comprehensive and robust assessment. During each run, we evaluated the models based on two key metrics: classification accuracy and loss.

✓ **Definition of the specific measure or statistics used to report results.**

- The average accuracy and error of the mean are calculated from all folds for the arousal score.
- The average accuracy and error of the mean are calculated from all folds for the valence score.
- Average loss per fold and error of the mean are calculated from all folds for the valence score.

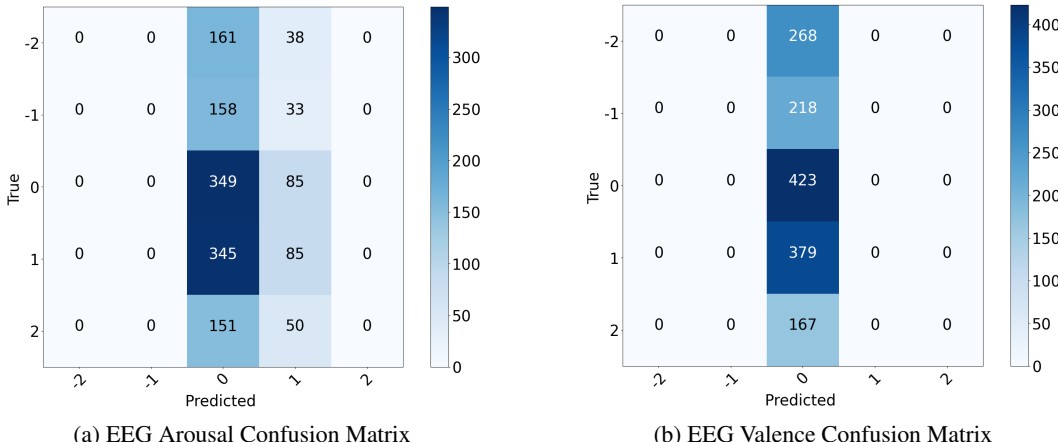

| (a) EEG Arousal Confusion Matrix | (b) EEG Valence Confusion Matrix |

Figure 4: Confusion matrices for arousal and valence prediction based on EEG data with a 1-second window and a 0-second offset.

 – Confusion matrix for valence score and arousal score.

These measures provide a clear understanding of the reported results, including accuracy and variability for both arousal score and valence score, as well as the average loss and its variability across different folds. Additionally, the confusion matrix provides insights into the classification performance for valence score and arousal score.

✓ **A description of results with central tendency (e.g.mean) & variation (e.g.errorbars).** For arousal score, the average accuracy is reported as the mean, indicating the central tendency of the data. The standard deviation is provided as a measure of variation, reflecting the spread or dispersion of the accuracy scores. This helps to understand the consistency or variability in the performance of the model for arousal score.

Similarly, for valence score, the average accuracy is presented as the mean, representing the central tendency. The standard deviation serves as a measure of variation, illustrating the extent of variability in the accuracy scores for valence score.

The average loss per fold provides the central tendency measure for the model's performance in terms of loss. It gives an insight into the average magnitude of errors made by the model across different folds. The standard deviation of the loss per fold helps to understand the variability or dispersion of the loss values, indicating how consistent or varied the model's performance is across different folds.

✓ **The average runtime for each result, or estimated energy cost.** The training of the fNIRS CNN model required 1.5 seconds for each fold of cross-validation, utilizing 24 GB of RAM and 2.5 GB of GPU memory. In contrast, the EEG model took 9 seconds per fold, consuming 35 GB of RAM and 7 GB of GPU memory.

✓ **A description of the computing infrastructure used.** The models were trained on high performance computing cluster with dual AMD EPYC 7542 32-Core processors (3.3 GHz), 1 TB of RAM and two Nvidia A100 GPUs.

## F.2 Correlation experiments

For all datasets used, check if you include:

✓ **Relevant statistics.** Tables 6 and 8 present the number of (group session, task) combinations considered and accepted for computing fNIRS correlations among participants, and tables 7 and 9 present the experiments rejected and the reasons for their rejections. Some experiments were not included in the final results because they do not have physio data recorded for one of the tasks. Among all experiments that were considered for the Ping Pong Cooperative task, one experiment was rejected because of its abnormally high task score.

Furthermore, unlike the physio task data synchronized dataset we discuss in § J.1 that provides 1000 Hz synchronized EEG-EKG-GSR data, our physio correlation analysis

Table 6: Experiments considered and accepted into the computation of results for fNIRS data.

| Task | Experiments accepted | Total # Experiments considered |
|------|---------------------|-------------------------------|
| Ping Pong Cooperative | 38 | 40 |
| Minecraft Saturn A | 36 | 36 |
| Minecraft Saturn B | 27 | 27 |

Table 7: Group sessions for which the data from the Ping Pong Cooperative task were rejected from the computation of results for fNIRS channels.

| Group session | Reason for Rejection |
|---------------|---------------------|
| `exp_2023_01_30_13` | Missing fNIRS data |
| `exp_2023_01_31_14` | Task score greater than 20 |

experiment uses synchronized EEG-EKG-GSR data synchronized at 500 Hz and fNIRS synchronized at 10 Hz.

- ✓ **Details of train/validation/test splits.** NA

- ✓ **An explanation of any data that were excluded, and all pre-processing steps.** Notably, there were three experiments that were not processed by both the physio synchronization program and the physio correlation analysis program. Two of these experiments contain Minecraft mission data that require additional cleaning to be useful. The last experiment contain file structure that also nseeds to be cleaned.

  During the physio correlation analysis process, three more experiments were chosen to be removed from the analysis. These experiments contain very high physio channel correlation values but their Minecraft scores were extremely low. These experiments were considered noise and were removed from the physio correlation analysis.

- ✓ **A link to a downloadable version of the dataset or simulation environment.** The intermediate dataset (chunked data) used for these experiments is available at `https://tomcat.ivilab.org/derived-data-products`.

- ✓ **For new data collected, a complete description of the data collection process, such as instructions to annotators and methods for quality control.** The data collection process is detailed in the main paper and in Appendix H.

For all shared code related to this work, check if you include:

- ✓ **Specification of dependencies.** The dependencies are listed in the `requirements.txt` file of the code respository.

- ✓ **Training code.** N/A

- ✓ **Evaluation code.** The evaluation code is included in the supplementary material. (`code/tomcat-signal-correlation-analysis-main`, `code/synchronize_signal_task`, and `code/signal_filtering`). Specifically, the evaluation metric is $R^2$, which evaluates how well linear regression predicts the team score from physio correlation.

- ✓ **(Pre-)trained model(s).** N/A

- ✓ **README file includes table of results accompanied by precise command to run to produce those results.** A README file is included in the code repository, but it does not include a table of results.

For all reported experimental results, check if you include:

- ✓ **The range of hyper-parameters considered, method to select the best hyper-parameter configuration, and specification of all hyper-parameters used to generate results.** NA

- ✓ **The exact number of training and evaluation runs.** The evaluation is ran once.

Table 8: Experiments considered and accepted into the computation of results for EEG data.

| Task | Experiments accepted | Total # of Experiments considered |
|---|---|---|
| Ping Pong Cooperative | 36 | 40 |
| Minecraft Saturn A | 33 | 35 |
| Minecraft Saturn B | 26 | 27 |

Table 9: Experiments rejected from the computation of results for EEG data.

| EEG with Task | Experiment Rejected | Reason for Rejection |
|---|---|---|
| Ping Pong Cooperative | exp_2022_10_24_12 | Missing EEG data |
| | exp_2023_01_30_13 | Missing EEG data |
| | exp_2023_01_31_14 | Task score greater than 20 |
| | exp_2023_05_01_13 | Missing EEG data |
| Minecraft Saturn A | exp_2023_05_01_13 | Missing EEG data |
| | exp_2023_05_02_14 | Missing EEG data |
| Minecraft Saturn B | exp_2023_05_02_14 | Missing EEG data |

✓ **A clear definition of the specific measure or statistics used to report results.** We evaluate the linear regression model using the R-square ($R^2$) metric, which indicates how well the the model can predict team score given physio correlation among participants.

✓ **Results with central tendency** The linear regression has low $R^2$ score overall.

   – $R^2$ of fNIRS for Ping Pong Cooperative linear regression: Mean 0.0915, Var 0.007
   – $R^2$ of fNIRS for Minecraft Saturn A linear regression: Mean 0.069, Var 0.010
   – $R^2$ of fNIRS for Minecraft Saturn B linear regression: Mean 0.059, Var 0.006
   – $R^2$ of EEG for Ping Pong Cooperative linear regression: Mean 0.060, Var 0.001
   – $R^2$ of EEG for Minecraft Saturn A linear regression: Mean 0.045, Var 0.002
   – $R^2$ of EEG for Minecraft Saturn B linear regression: Mean 0.058, Var 0.008

✓ **The average runtime for each result.** To synchronize the data with the computing infrastructure below, the synchronization of physio data and task data took one hour to complete. Then, from the output of the physio synchronization program, the physio correlation analysis took under 10 minutes to complete.

✓ **The computing infrastructure used.** The models were trained on a server with dual AMD EPYC 7542 32-Core processors, 1 TB of RAM and dual Nvidia A100 GPUs.

# G   Author statement

The authors bear all responsibility in case of any violation of rights during the collection of the data or other work, and will take appropriate action when needed, e.g., to remove data with such issues.

# H   Data collection

## H.1   Questionnaires

Questionnaire data were collected and managed using REDCap electronic data capture tools hosted at the University of Arizona [134–136]. REDCap (Research Electronic Data Capture) is a secure, web-based software platform designed to support data capture for research studies, providing 1) an intuitive interface for validated data capture; 2) audit trails for tracking data manipulation and export procedures; 3) automated export procedures for seamless data downloads to common statistical packages; and 4) procedures for data integration and interoperability with external sources.

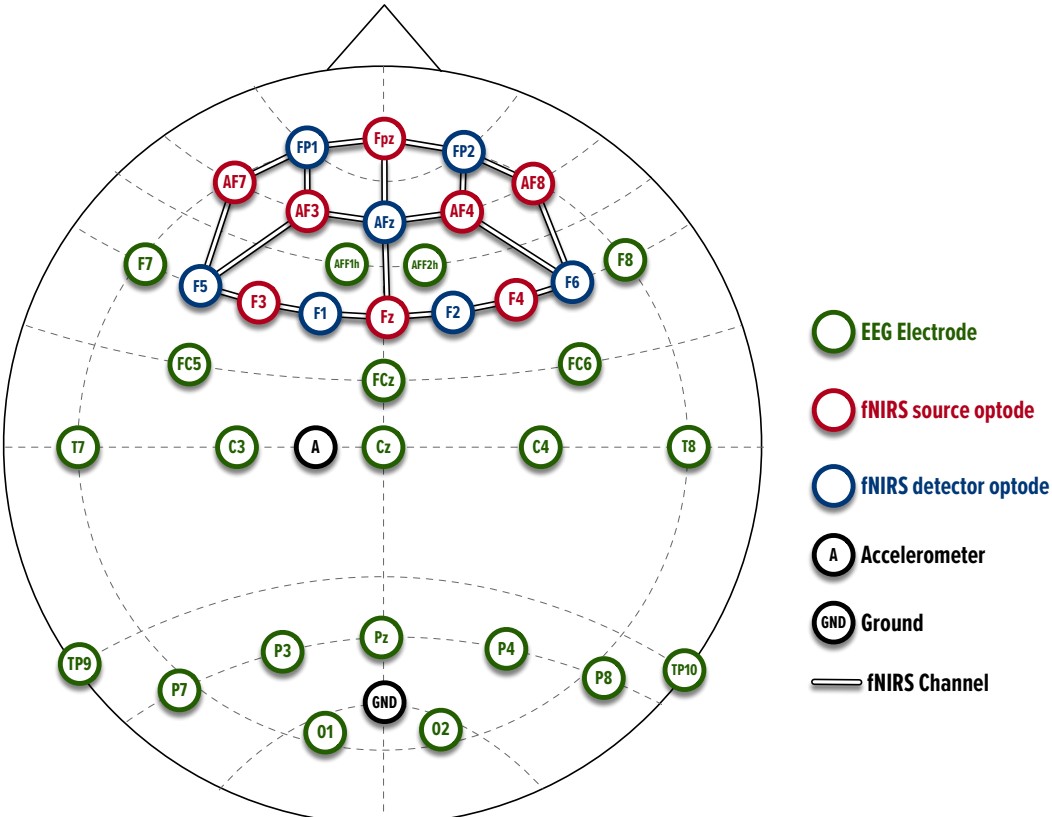

Figure 5: The montage we used for combined EEG/fNIRS data acquisition. EEG electrodes were located in the anterior frontal (AFF1h, AFF2h), frontal (F7, F8), frontocentral (FC5, FCz, FC6), central (C3, Cz, C4), occipital (O1, O2), temporal (T7, T8) and parietal (P7, P3, Pz, P4, P8) regions. fNIRS optodes were located in frontal (Fz, F1, F2, F3, F4, F5, F6), anterior frontal (AF3, AF4, AFz, AF7, AF8), frontal polar (FP1, FPz, FP2). The line is the channel formed when fNIRS source optode (red) and fNIRS detector (blue) optode are combined.

## H.2 EEG

**Equipment** We used actiCHamp Plus (Brain Vision, LLC), which records and streams EEG signals at 500 Hz.

**Montage** We used an adapted, 32-electrodes, 10-20 electrode placement system [137], to provide adequate coverage of all regions of the brain [54]. The actual placement of the electrodes (i.e., the montage) used for our experiment is shown in the combined EEG/fNIRS montage in Figure 5.

For experiments prior to November 22, 2022, our dataset contains the following EEG channels: AFF1h, F7, FC5, C3, T7, TP9, Pz, P3, P7, O1, O2, P8, P4, TP10, Cz, C4, T8, FC6, FCz, F8, AFF2h, AFF5h, FC1, CP5, CP1, PO9, Oz, PO10, CP6,CP2, FC2, AFF6h. For experiments after November 22, 2022, our dataset contains the following EEG channels: AFF1h, F7, FC5, C3, T7, TP9, Pz, P3, P7, O1, O2, P8, P4, TP10, Cz, C4, T8, FC6, FCz, F8, AFF2h (omitting AFF5h, FC1, CP5, CP1, PO9, Oz, PO10, CP6,CP2, FC2, AFF6h) to meet the time constraint of each experiment.

**Software** EEG data was streamed using LSL-actiCHamp [138], streamed over the local area network using Lab Streaming Layer [139] and recorded using LabRecorder [140] to XDF files [141].

## H.3 fNIRS

**Equipment** We used NIRSport2 (NIRx Medical Technology LLC), which records and streams fNIRS signals at 10.2Hz.

**Software**  fNIRS data was streamed using Aurora fNIRS [142], streamed over the local area network using Lab Streaming Layer (LSL) [139] and recorded using LabRecorder [140] to XDF files.

**Montage**  See Figure 5.

## H.4  EKG

We measured heart activity continuously throughout the experiment. We connected EKG leads to the Aux channel of the EEG/fNIRS caps. The EKG data was streamed alongside EEG and GSR data using LSL and written to XDF files.

## H.5  GSR

GSR was recorded continuously throughout the experiment at a sampling rate of 500 Hz. Two bipolar electrodes were placed on the ulnar border of the palm on the participants' right hand. We used an electrode paste that mimicked the salt concentration of sweat (i.e., 0.5% saline) to improve signal quality. Participants' hands were wrapped in medical tape to secure the electrodes. Leads connected the electrodes to the Aux channel of the EEG/fNIRS cap. The GSR data was streamed alongside EEG and EKG data using LSL and written to XDF files.

## H.6  Eye tracking

**Equipment**  The eye tracking system used in this study was Pupil Core (Pupil Labs GmbH). The eye tracker was connected to a computer via USB and eye tracking data was streamed using LSL at 250 Hz and written to XDF files.

**Calibration**  The eye tracker is positioned in front of the participant's eyes, then eye tracker was calibrated using a 5-point calibration procedure (the participants were asked to look at the target on the screen without moving their head). The calibration procedure gave a confidence score and we made sure it was over 90%.

# I  Data Extraction and Labeling

The data extraction process encompassed retrieving diverse stream types such as EEG, EKG, GSR, fNIRS, and Gaze from a consolidated XDF file, which had been stored using LSL. For enhanced analysis and processing capabilities, each stream was then extracted from the xdf file and converted into separate Python dataframes utilizing pyXDF [143].

To label the stream data frames with task-specific labels, the start and stop times of each task were obtained from timestamps assigned by the task itself. For instance, let us consider the rest state. Its start and stop timestamps were compared with the timestamps in the EEG data frame. Whenever a match was found, the corresponding section in the EEG data was labeled as "start_rest_state" and "stop_rest_state" accordingly. This process was repeated for the remaining streams, associating the start and stop times of each task with their respective sections in the corresponding data frames. By labeling the data in this manner, it becomes easier to identify and analyze specific segments of interest within the recorded streams.

The extracted and labeled dataframes can be either saved as a CSV file or forwarded to the subsequent stage of processing, known as signal processing, which will be described in the following section.

Note:

- For fNIRS dataframes, they consist of various components including raw signals representing changes in attenuation values detected from two wavelengths (W1: 760 nm and W2: 850 nm), as well as HbO (oxygenated hemoglobin) and HbR (deoxygenated hemoglobin) signals. When saving the data as a CSV file, typically only the HbO and HbR signals are preserved. However, if further signal processing is required, we retain both the raw signals and the HbO and HbR signals for subsequent analysis.

- For EEG, EKG, and GSR dataframes, they were converted to voltage values since the data was recorded in microvolts. The conversion was performed by multiplying the data by the corresponding scale factor of $1 \times 10^{-6}$.

## J  Signal Processing

### J.1  Synchronization of EEG, EKG, GSR, and fNIRS Signals

In multimodal neuroimaging studies, synchronizing signals from multiple modalities is a crucial step to conducting comprehensive studies on all these modalities together. The discrepancy between EEG, EKG, GSR, and fNIRS signals' time series is an issue that researchers encounter frequently due to the limitations of recording hardware or the necessity to remove invalid signals. Additionally, these two modalities have distinct recording rates, further complicating their alignment. To facilitate comprehensive evaluation of EEG, EKG, GSR, and fNIRS signals, it is essential to synchronize these signal modalities.

The synchronization process is a two-step approach involving noise artifacts removal of EEG and fNIRS, followed by the synchronization process of the signals to the desired sampling rate,

#### J.1.1  Removing Noise in EEG Signals with Notch Filter

EEG signals often exhibit susceptibility to artifacts, an interference that can be attributed to several sources. For instance, physiological factors such as eye movements or blinks can induce such artifacts [125], as can environmental elements like fluorescent lighting or grounding complications [126].

Upon thorough examination and visualization of the raw EEG data, we identified a consistent 60 Hz electrical disturbance within the signal, along with corresponding harmonics. An anomalous peak was also noted around the 5 Hz mark, potentially attributable to a grounding irregularity or an other environmental factors.

With the aid of MNE-Python [127], we efficiently mitigated these intrusive noises by deploying a notch filter. The filter was configured with a frequency of 60 Hz, a transition bandwidth of 9 Hz, and notch widths of 2 Hz. The filter is applied to each channel separately.

#### J.1.2  Mitigating Artifacts in fNIRS Signals Utilizing Bandpass Filter

fNIRS signals are often susceptible to motion artifacts (MA) stemming from physiological activities, including cardiac and respiratory disturbances. These artifacts become particularly noticeable in the measurement of oxyhemoglobin (HbO) and deoxyhemoglobin (HbR) concentrations within the signal channels.

To address these challenges, we employed a bandpass filter as an effective noise reduction strategy. The filter was calibrated in line with the recommendations provided by [128]. With a low cutoff bandwidth of 0.01 Hz and a high cutoff bandwidth of 0.2 Hz for the 4$^{\text{th}}$ order Butterworth method, the filter was tailored to selectively allow signal components within this frequency range while attenuating components outside the range. The filter is applied to each channel separately.

#### J.1.3  Pre-processing EKG and GSR Signals

To remove noise and improve peak-detection accuracy for EKG signals, we employed a finite impulse response (FIR) filter with 0.67 Hz low cutoff frequency, 45 Hz high cutoff frequency, and order of $1.5 \times$ sampling rate (where sampling rate is 500 Hz) implemented by NeuroKit2 [144] on top of the BioSPPy package.

We removed noise and smoothed the GSR signals using a low-pass filter with a 3 Hz cutoff frequency and a 4$^{\text{th}}$ order Butterworth filter, both implemented by Neurokit2.

#### J.1.4  Synchronization of EEG, EKG, GSR, and fNIRS Signals

To obtain the EEG, EKG, GSR, and fNIRS signals at the desired sampling rate, we resampled the signals using the FFT-based resampling method `mne.filter.resample` available in the Python

MNE library [127]. Then, we used linear interpolation to synchronize all the signals to a time series with a regular interval matching the desired sampling rate.

## J.2 Synchronizing Task Data with EEG, EKG, GSR, and fNIRS Resampled Signals

Understanding the relationship between participants' behaviors, environmental stimuli, and neuroimaging data requires a precise synchronization of task data with the corresponding EEG, EKG, GSR, and fNIRS signals. By aligning these data streams, we can examine the influence of environmental stimuli on the participants' neuroimaging signals, which in turn, impact their behavior and task performance.

The process of integrating EEG, EKG, GSR, and fNIRS signals with task data starts with grouping of signals by the tasks during which they were recorded, followed by the synchronization of the task data to the corresponding EEG, EKG, GSR, and fNIRS signals.

### J.2.1 Grouping EEG, EKG, GSR, and fNIRS Signals by Task

The preliminary step in our approach to synchronizing EEG, EKG, GSR, and fNIRS signals with the task data involves the grouping of EEG, EKG, GSR, and fNIRS signals by the tasks during which the signals were recorded. The task data can be categorized into two distinct types: status-based and event-based data.

**Status-based task data**    This type of task data represent the current state of the task, such as task score. For each task, the grouping process of these data begins by including the EEG, EKG, GSR, and fNIRS signal recorded immediately before the task initiation and immediately following task completion. This ensures no data is overlooked at the boundaries of the task. Subsequently, all EEG, EKG, GSR, and fNIRS signals recorded between these two points are included, forming a complete set of signals associated with the task.

**Event-based task data**    This type of task data, on the other hand, correspond to specific events that occur during the task, such as affective task arousal or the submission of a valence score. For each task, we determine the EEG, EKG, GSR, and fNIRS entry associated with the first event and the last event. These signal entries, as well as all entries recorded between these points, are included into the data set related to the task.

### J.2.2 Synchronizing Task Data with EEG, EKG, GSR, and fNIRS Signals

Having grouped the EEG, EKG, GSR, and fNIRS signals according to task type, we then proceed to synchronize these signal entries with their respective task data.

**Status-based task data**    The synchronization is accomplished by assigning the status data recorded closest in time to each EEG, EKG, GSR, and fNIRS signal entry. This method ensures that each EEG, EKG, GSR, and fNIRS entry is paired with the most representative status data.

**Event-based task data**    We assign each event data to the EEG, EKG, GSR, and fNIRS signal entry recorded at the time closest to the occurrence of the event. Those EEG, EKG, GSR, and fNIRS signal entries without a corresponding event data are left unassigned, signifying that no specific event occurred during these recordings.

## K  Speech Elicitation Task

### K.1  Reading Task

In the reading task, participants were shown a short piece of text describing the Minecraft environment used in the experiment and instructed to read it aloud with a natural tone and pace. The purpose of this task was to elicit an identical set of specialized Minecraft terminology that the participants were likely to use during the experiment, in order to control for possible word-frequency and other linguistic effects. This task also creates a fallback audio recording in case participants are not sufficiently

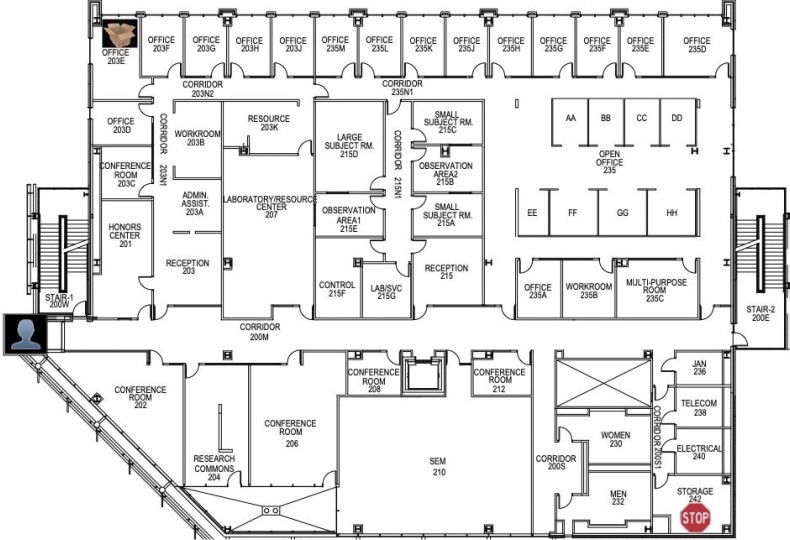

Figure 6: Top-down view of a building floor. Each room has a unique label. The map also contains three places of interest, marked with overlaid graphics.

verbose during the spontaneous elicitation (i.e., during the Minecraft tasks), and gives them a short introduction to the Minecraft environment.

Participants read out the following description of the Minecraft environment and game objectives:

Minecraft is a sandbox construction game in which players can control the environment. The Minecraft world for this game will require players to pick roles. You are tasked with search and rescue in this scenario. Keep the channels of communication open. Since this is a rescue site, be careful as you clear rubble and identify the critical victims.

Some of the tools you will have access to include a medkit, a stretcher, and a hammer. Using the map, identify the zone the team wishes to work on, and proceed to clear the floor and the rooms. As you orient yourself in the environment, pay attention to doors and windows.

Player roles include a medic, and engineer and a searcher. You can pick your roles based on the game's requirements. Pay attention to the clock. Keep the game's objectives in sight as you proceed.

## K.2   Map Task

Next, we used a map task in order to capture audio containing speech representative of the participant's typical speaking style. Participants were shown a building floor plan with several labelled regions and icons indicating locations of interest (see Figure 6).

Participants were instructed to imagine a friend standing at one of these locations and provide descriptive verbal instructions directing their friend to the other two locations, using as many details from the map and its labels as they could. This task also served as a way for participants to practice giving navigational guidance to their teammates prior to doing so in the Minecraft tasks.

In the instructions, they were informed that their friend was located at the building entrance on the mid-left of the map(indicated by the person icon) and needed to first travel to an office located in room 203 E (top left, box icon) and then to Storage 242 (bottom-right, indicated with a 'stop' icon). The instructions were as follows:

> For the following map task, you will be helping your friend reach their destination on a building floor. They are located on the entrance on the left side of the map, at the point with the person emoji. Your friend does not have access to the map. Therefore, you will need to give them directions so that they can:
>
> > Reach the room with the cardboard box (top left).
> > Travel to the room with the stop sign (bottom right).
>
> Please make sure that your friend has as much information and details as possible so that they can navigate the floor without getting lost.

## L  IAPS Stimuli for the affective task

For the affective task, we used stimuli from the International Affective Picture System (IAPS) [145] following Balconi, Grippa, and Vanutelli [86]. Stimuli were selected from the neutral, pleasant (high and low), and unpleasant (high and low) categories.

The IAPS stimuli numbers chosen are listed below.

- *Individual task*: 1525, 2025, 2352.2, 2487, 2521, 2635, 3019, 5260, 5900, 7270, 7405, 8034, 8485, 8531, and 9910.
- *Team task*: 1019, 1710, 1947, 2208, 2683, 2703, 3005.2, 5621, 6415, 6930, 7021, 7360, 7484, 8490, and 8502.

## M  Limitations

We note the following limitations.

The following factors limit the generalizability of the dataset:

- The dataset is primarily comprised of English-speaking undergraduates at a university in the US.
- Minecraft is known for low visual fidelity, so it is possible that results from these experiments may not generalize well to real-life SAR scenarios.

Additionally, due to the complexities of collecting this kind of data, there are a fair number of instances of missing data for a subset of modalities for a subset of sessions.

Finally, while we provide an initial release of the data with this paper, we have not been able to provide some of the planned modalities and documentation by the camera-ready deadline. However, we will continue to work on providing these at a rapid pace, and expect to have all the planned data modalities and documentation available by the time of the NeurIPS 2023 conference.

## N  Affect prediction score changed in camera-ready manuscript

The results for the experiment on affect prediction in this camera-ready version of the manuscript are significantly different (i.e., the accuracy is lower) from the results in the accepted version of the paper (i.e., after the rebuttal period). This is because our intention was to test on individual-image pairs not seen in training, which is very challenging. However, there was a bug in the splits, where information was leaked from training to testing.

The results in this camera-ready manuscript are generated by code that incorporates a fix for the data leakage.

## O   Consent form

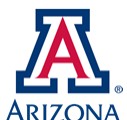

# University of Arizona
# Consent to Participate in Research

**Study Title:** ToMCAT: Theory of Mind-based Cognitive Architecture for Teams

**Principal Investigator:** Adarash Pyarelal

**Sponsor:** This research is funded by the U.S. Department of Defense.

### Summary of the research

**This is a consent form for participation in a research project.** Your participation in this research study is voluntary. It contains important information about this study and what to expect if you decide to participate.  Please consider the information carefully. Feel free to ask questions before making your decision whether or not to participate.

The purpose of this research is to develop an artificially intelligent computer agent that can help human teams perform better. If you choose to participate, you will play a Minecraft game, while the computer agent watches and sometimes provides advice. During the game we will measure your brain activity, eye gaze, heart rate, muscle activity and sweating and the computer agent may ask you questions about how you are feeling. We will also videotape you while you play the game. Your participation in the study will take place during two lab sessions and will require about 3.5 hours of your time. There is some risk that someone could recognize you in the videotape, or that you will be upset by the game. There is no direct benefit to you of participating.

### Why is this study being done?

This research study is being done so that we can develop an artificially intelligent (AI) agent that can participate in human teams to improve the team's performance. The first goal of the research is to train the agent to observe the team (including video of the team, chat messages, brain activity, heart rate, sweating and eye gaze) and predict from those observations what the humans are trying to do in the game, how coordinated they are as a team, and how they feel about the game and about each other. The second goal of the research is to train the agent to communicate with the humans and take actions that could improve the team's performance.

### What will happen if I take part in this study?

If you choose to participate you will do the activities listed below during two lab sessions.

- At the pre-session, you will be asked to answer some brief socio-demographic questions (e.g., your age, ethnicity, etc.) and do a speech/language task where you will read some sentences out loud and provide spoken directions for someone else to navigate from a start to an end point on a map that will be given to you. You will then be asked to schedule a time within 1-3 weeks to complete the second testing session.
- At the testing session, sensors will be placed on your torso and head that will measure heart rate, muscle activity, sweating, brain activity and eye gaze. You will also wear a headset with speakers and a microphone.

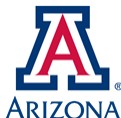

- The session will be videotaped.
- You will be asked to complete the following three baseline tasks on the computer: finger tapping, looking at emotional pictures, and a ping-pong game.
- After the baseline tasks you will play a game in Minecraft. You will be given a tutorial about how to play the game and a chance to practice the game.
- You will then complete a 5-minute competency test so that we can better understand video-gaming proficiency across individuals.
- The Minecraft game will either involve: 1) saving villagers, who are hiding in closed rooms within a building, from various creatures that appear in Minecraft (e.g., zombies, creepers, etc.), or 2) a search and rescue mission where you must navigate through a building and rescue victims and remove hazards. There will be a 20-minute time limit on the game. Your performance during the game will be recorded.
- Every 3 minutes, the computer agent may pause the game and ask you to answer some questions about what you are thinking and feeling. The agent may also offer you some information or advice. The agent may be acting in a truly "intelligent" way, coming up with its own information and advice, or it may be preprogrammed or controlled by the research assistant.
- After the game, all sensors and the headsets will be removed. You will answer another set of questions about your feelings about the game and the agent.
- Finally, the game may be played back on the computer screen and you may be asked to pause the play-back (by pushing a computer key) at any point that you remember having a plan in mind (e.g., an idea about what you were trying to do, or a plan about a sequence of actions that you intended to take), or changing your plans, or feeling confused about what to do. You will be asked to tell the research assistant what you remember, and the research assistant may ask clarifying questions. This will be audio-recorded and may be translated into text using natural language processing algorithms.

### How long will I be in the study?
You will be done with the study after the half hour pre-session and the three-hour lab session, both of which will occur sometime in the next few weeks.

### How many people will take part in this study?
Approximately 900 people will take part in this study.

### Can I stop being in the study?
**Your participation is voluntary.** You do not need to participate in this study. If you decide to take part in the study, you may leave the study at any time. No matter what decision you make, there will be no penalty to you, and you will not lose any of your usual benefits. Your decision will not affect your future relationship with The University of Arizona. If you are a student or employee at the University of Arizona, your decision will not affect your grades or employment status.

### What risks or benefits can I expect from being in the study?
The only risks are that you may find the game to be stressful. While playing the game, it is also possible that you may experience some motion sickness, but we have measures in place should that

occur. There is also a minimal risk that your data, including the videotape, could be accessed by someone not on the research team, although we will be very careful to keep all the data, including the video secure and confidential.

There are no benefits to you for participating, except that you may find it fun to play the game.

### Will I be paid for participating in the study or experience any costs?

Participants will receive a $10.00 Amazon gift card for each hour of the study they complete. You will attend the half hour pre-session and one 3-hour lab session, so you will receive a $35.00 Amazon gift card at the end of the lab session. If you cannot, or choose not to, participate in both sessions you will still receive gift cards for the session you do attend.

You also have a chance of winning an additional $20.00 Amazon gift card if you get the highest score on the video game out of 20 lab sessions.

You also have the option of receiving research credit if you are a student participating for credit through the UA SONA system in the Department of Psychology. Participants will attend the half hour pre-session and one three-hour lab session, so you will receive 3.5 to 4.0 SONA credits at the end of the lab sessions. If you cannot, or choose not to, participate in all sessions, you will still receive the SONA credits for the sessions you do attend.

Compensation for participation in a research study is considered taxable income for you. If your compensation for this research study or a combination of research studies is $600 or more in a calendar year (January to December), you will receive an IRS Form 1099 to report on your taxes. Please note, if you are an employee of UArizona, any compensation from a research study is considerable taxable income.

The only costs to you for participating are your time and any transportation costs due to attending the lab session(s).

### Will my study-related information be kept confidential?

Every effort will be made to keep your information confidential. All the data for the study will be encrypted and password protected. Most of the measures that you provide in the study will be anonymous. You will be given a random ID number and your name will not be recorded anywhere. The audio and video recordings of the sessions are the only measures that could be used to identify you. You will not be identified in any report or publication of this study. However, you could be identified from the videotape of your session, but the video will not show anything sensitive. It will just show your face and torso as you play the video game.

Any audio data will be made publicly available but no personal information will be attached to any of the audio files. The audio recordings will include anything you or the other participants say during the speech/language tasks at presession as well as the planning and playing stages of the mission. We will ask you and the other participants to not speak your names or the names of other participants to

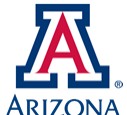

maintain confidentiality; instead, you may call each other by your computer's name or any other anonymous manner. You can review the recording and/or request your recording be removed by contacting the principal investigator of the study (see contact information at the end of this consent form); otherwise, the audio recordings will be kept indefinitely.

We will also be collecting and storing data through REDCap (Research Electronic Data Capture). The REDCap electronic data management (EDM) system at the University of Arizona is housed on 2 virtual servers; one supporting database services and the other web services. In REDCap, appropriate measures are in place to maintain confidentiality and security of all incoming data.

The information that you provide in the study will be handled confidentially. However, there may be circumstances where this information must be released or shared as required by law. The University of Arizona Institutional Review Board; other federal, state, or international regulatory agencies; or the sponsor of the study, if any, may review the research records for monitoring purposes. In particular, representatives of the U.S. Department of Defense will have access to research records as part of their responsibilities for human subjects protection oversight of the study.

### Will my study-related information be used for future research?

Information that may identify you, such as the videotape of the session, may be used for future research without additional consent. All data from the study will be kept forever and cannot be withdrawn once it has been collected, with the exception of the audio recordings as explained above.

Your data will also be shared with other research teams collaborating on this study at other universities and research companies. Arizona State University is organizing a secure data base that we and 12 other research teams will use to share data, including the videos from this study. Your data will be included in this data base and may be used by other researchers in future studies without contacting you.

### Who can answer my questions about the study?

For questions, concerns, or complaints about the study you may contact **Dr. Adarsh Pyarelal via phone at (503)-360-8824, or via e-mail at adarsh@arizona.edu.**

**For questions about your rights as a participant in this study or to discuss other study-related concerns or complaints with someone who is not part of the research team, you may contact the Human Subjects Protection Program Director at 520-626-8630 or online at https://research.arizona.edu/compliance/human-subjects-protection-program.**

**Signing the consent form**

I have read (or someone has read to me) this form, and I am aware that I am being asked to participate in a research study.  I have had the opportunity to ask questions and have had them answered to my satisfaction.  I voluntarily agree to participate in this study.

I am not giving up any legal rights by signing this form.  I will be given a copy of this form.

______________________________        ______________________________________        ________________
**Printed name of subject**                          **Signature of subject**                                              **Date**

# P  Recruitment materials

## RESEARCH VOLUNTEERS NEEDED!

**The Temporal Interpersonal Emotion Systems (TIES) lab in Family Studies and Human Development at the University of Arizona is searching for volunteers who are:**

1.  18+ years old

2.  Willing to spend up to three hours in the lab interacting with a virtual agent and other human teammates to complete a task in a video game environment.

3, No prior gaming experience required.

Participants will be paid for their time.

**Please e-mail HIS.FSHD@GMAIL.COM, or text/call at (520)-497-0937 if you are interested.**

Teamwork study
–TTIES Group
(520)-497-0937
HIS.FSHD@GMAIL.COM

Teamwork study  –
TIES Group
(520)-497-0937
HIS.FSHD@GMAIL.COM

Teamwork study
–TIES Group
(520)-497-0937
HIS.FSHD@GMAIL.COM

Teamwork study
– TIES Group
(520)-497-0937
HIS.FSHD@GMAIL.COM

Teamwork study
–TIES Group
(520)-497-0937
HIS.FSHD@GMAIL.COM

Teamwork study
–TIES Group
(520)-497-0937
HIS.FSHD@GMAIL.COM

Teamwork study
–TIES Group
(520)-497-0937
HIS.FSHD@GMAIL.COM

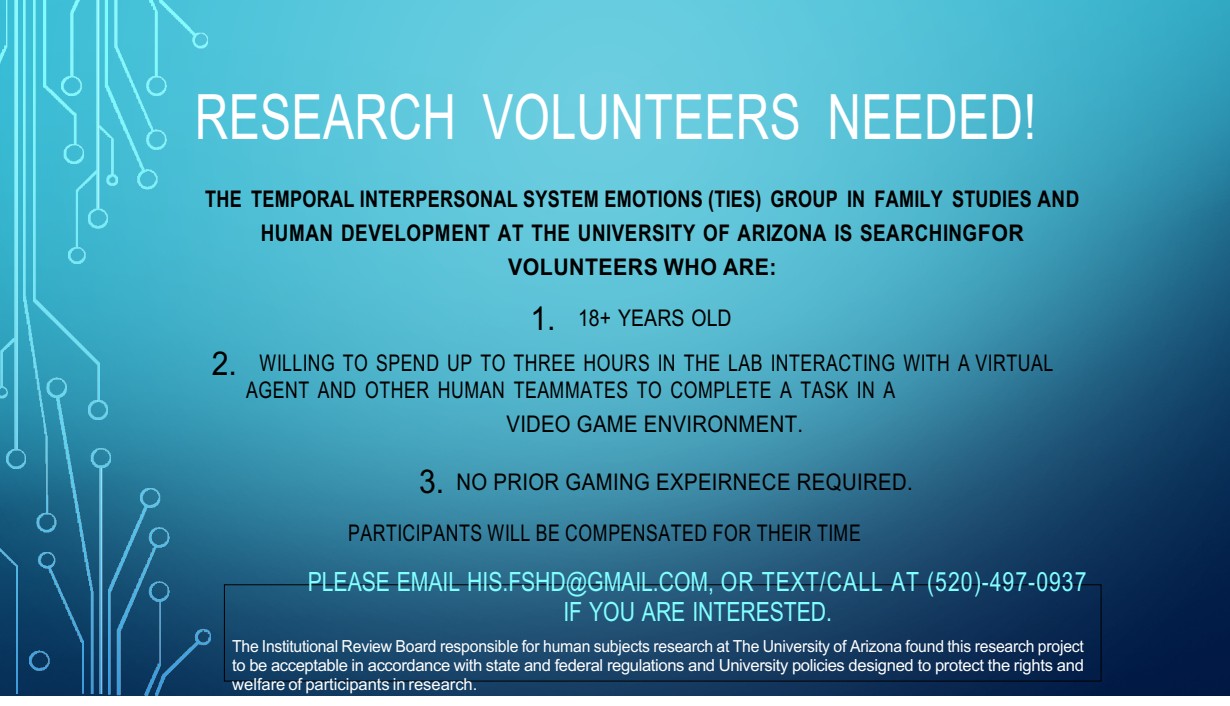

# RESEARCH VOLUNTEERS NEEDED!

**THE TEMPORAL INTERPERSONAL SYSTEM EMOTIONS (TIES) GROUP IN FAMILY STUDIES AND HUMAN DEVELOPMENT AT THE UNIVERSITY OF ARIZONA IS SEARCHINGFOR VOLUNTEERS WHO ARE:**

1. 18+ YEARS OLD

2. WILLING TO SPEND UP TO THREE HOURS IN THE LAB INTERACTING WITH A VIRTUAL AGENT AND OTHER HUMAN TEAMMATES TO COMPLETE A TASK IN A VIDEO GAME ENVIRONMENT.

3. NO PRIOR GAMING EXPEIRNECE REQUIRED.

PARTICIPANTS WILL BE COMPENSATED FOR THEIR TIME

PLEASE EMAIL HIS.FSHD@GMAIL.COM, OR TEXT/CALL AT (520)-497-0937 IF YOU ARE INTERESTED.

**Extra credit**

You have two extra credit opportunities in this course.  Extra credit opportunities are optional.

You can choose either Option 1 **OR** option 2 to earn up to_____points. **Either of these options will take about the same amount of effort. The maximum number of extra credit points you can earn on Option 1 or Option 2 is 10 points.**

For Option 1 (online survey) **OR** Option 2 (paper): Deadline is by the time class starts on the date specified in the Course Outline.

**Option 1:**
Participate in a study examining how individuals and teams work with a virtual agent to complete a task in a video game environment.
Inclusion criteria:
   1)  At least 18 years of age
   2)  Willingness to be in the lab for up to 3 hours
Exclusion criteria:
   1) No major physical limitations that would interfere with completing tasks on a computer (e.g., limited vision or hearing, problems with fine motor control).

*To obtain extra credit:* The individual who will participate in the study will need to register on the SONA system. The researcher will compile this information and return it to your instructor. Without this information, you will not be able to earn extra credit.

*Additional information and answers to frequently asked questions, **as well as a digital copy of the recruitment flyer, will soon be posted on D2L under Content under Extra Credit**. Once posted, please read this document before taking the survey, as questions you have should be answered in this document.*

**Option 2: The details of this option will be decided by the class instructor. Here is an example of the type of option that might be provided:**

For the second option you can write a paper. For this paper you should find an empirical article that uses one of the theories covered in class, and write a summary (two pages; single spaced; 1 inch margins; 12 point font) describing the article and discussing how it is relevant to the theory covered in class. Full points will only be given if the main points of the tenets are used correctly when explaining the empirical article. Include a copy of the article on which the review is based.

Include both your write up and the pdf of the article under dropbox on d2l as follows:

Word document (your write up) should be labeled as follows: Your last name. Your first name. Last name of author from article and year (e.g., 1. Smith. Bob. Sassler 2004)
For the actual article, please specify 2 and then switch the order (e.g., 2. Sassler 2004 Smith. Bob). This way, we will be able to match your write up with your article in the drop box. Including only the write up without the pdf will result in partial points for extra credit.

**Recruiting Scripts**

For social media and listservs

The Temporal Interpersonal Emotion Systems (TIES) research group, in the Department of Family Studies and Human Development, is currently recruiting participants for a study that will develop an artificially intelligent computer agent that can help human teams perform better. Participants will play a video game, either alone or with several other people, while the computer agent watches and sometimes provides advice. During the game we will measure the human player's brain activity to help the agent learn how to predict what people are thinking and feeling. All participants should be at least 18 years of age or older. Participants will be financially compensated for their time. If you are interested or have any questions, please email us at his.fshd@gmail.com or text/call us at (520)-497-0937.

or

The Institutional Review Board responsible for human subjects research at The University of Arizona found this research project to be acceptable in accordance with state and federal regulations and University policies designed to protect the rights and welfare of participants in research. Investigator: Adarsh Pyarelal, Ph.D.

For SONA

The Temporal Interpersonal Emotion Systems (TIES) research group, in the department of Family Studies and Human Development, is currently recruiting participants for a study that will develop an artificially intelligent computer agent that can help human teams perform better. Participants will play a video game, either alone or with several other people, while the computer agent watches and sometimes provides advice. During the game we will measure thehuman player's brain activity to help the agent learn how to predict what people are thinking and feeling. All participants should be at least 18 years of age or older. Participants can either be financially compensated for their time or receive extra course credit. There is also an alternative extra credit assignment available for those who are not eligible to participate. If you are interested or have any questions, please email us at his.fshd@gmail.com or text/call us at (520)-497-0937.

The Institutional Review Board responsible for human subjects research at The University of Arizona found this research project to be acceptable in accordance with state and federal regulations and University policies designed to protect the rights and welfare of participants in research. Investigator: Adarsh Pyarelal, Ph.D.

In-person presentations in undergraduate classes:

"Hello students, I am_________, a graduate student in the Department of Family Studies and Human Development. We are recruiting participants for a study that will develop an artificially intelligent computer agent that can help human teams perform better. Participants will play a video game, either alone or with several other people, while the computer agent watches and sometimes provides advice. During the game we will measure the human player's brain activity

to help the agent learn how to predict what people are thinking and feeling. All participants should be at least 18 years of age or older. Participants can either be financially compensated for their time or receive extra course credit. There is also an alternative extra credit assignment available for those who are not eligible to participate. If you are interested or have more questions, please email us at his.fshd@gmail.com.

The Institutional Review Board responsible for human subjects research at The University of Arizona has approved this study in accordance with state and federal regulations and University policies designed to protect the rights and welfare of participants in research. The primary investigator is Adarsh Pyarelal, Ph.D. This flyer will be provided on D2L. Thanks!"

In-person informal conversations:

" My lab is recruiting participants for a study that will develop an artificially intelligent computer agent that can help human teams perform better. Participants will play a video game, either alone or with several other people, while the computer agent watches and sometimes provides advice. During the game we will measure the human player's brain activity to help the agent learn how to predict what people are thinking and feeling. The Institutional Review Board responsible for human subjects research at The University of Arizona has approved this study in accordance with state and federal regulations and University policies designed to protect the rights and welfare of participants in research. The primary investigator is Adarsh Pyarelal, Ph.D. You can let me know if you're interested, or if you know of someone else who might be. "

