# OpenReview forum: "The ToMCAT Dataset"
_NeurIPS.cc/2023/Track/Datasets_and_Benchmarks — NeurIPS 2023 Datasets and Benchmarks Poster_

### Official Review · Reviewer_69jK · 2023-07-18
**Multi-modal database for the study of human-machine teaming**

**Rating:** 6
**Confidence:** 2
**Clarity:** Yes, well written.

**Strengths:**

The motivation is clear and the claim that the data should be useful to both computer and social scientists is convincing. The description of the data and collection is comprehensive and the data seems to be extremely rich.

The authors have made the data available to others via different routes, including both raw data and processed, derived data.

**Additional Feedback:**

None

**Correctness:**

Seems to be good, although I'm uncertain as to how realistic or representative is the artificial data collection scenario.

**Documentation:**

Sufficiently clear

**Ethics:**

No concerns - the authors state that the data contributors signed a consent form and that the data is appropriately anonymised.

**Limitations:**

Again, this area is one that I'm not familiar with, but the results don't seem to be so fantastic. They are described as being 'better than chance' and I find some appropriate level of discussion of this seemingly rather negative result to be lacking. The last paragraph in Section 7 seems to suggest that the authors do not have an 'understanding (of) what is going on'. This is hardly encouraging. I won't let this affect my recommendation too strongly since I understand that the experiments are only examples of how the data can be used.

**Opportunities For Improvement:**

Perhaps it is because this area of research is quite outside that of my own but, even if I did understand that they are supposed to provide example analyses only, I struggled to see the *link* between the exploratory experiments and, not just the data collection and intended use case scenario, but even the data itself - in other words, I didn't understand *how* the dataset used for experiments, nor how it was annotated beforehand, even where the ground-truth labels come from.

**Relation To Prior Work:**

Clear - described in the introduction and Section 2.

**Summary And Contributions:**

The paper presents a database for the study of human-machine teaming and contains multi-modal data collected in a number of different collaborative task scenarios.

---

> ### Author Response · Authors · 2023-08-22
> **Response to reviewer 69jK**
>
> Thank you for taking the time to review our paper, and for your valuable
> feedback! We are glad that you found our motivation clear and our documentation
> comprehensive. In this response, we will attempt to address the concerns you
> raised.
>
> > I struggled to see the link between the exploratory experiments and, not just
> the data collection and intended use case scenario, but even the data itself -
> in other words, I didn't understand how the dataset used for experiments, nor
> how it was annotated beforehand, even where the ground-truth labels come from
>
> Our apologies for not wording these descriptions more clearly. The fact is that
> this is an extremely large and complex dataset, so it is challenging to fit all
> the details within the NeurIPS page limit. But it is on us to do better. We
> have significantly expanded the Related work section, which might help
> contextualize the dataset better, and link it more strongly to the intended use
> case (i.e., research on human-machine teaming and artificial
> social intelligence).
>
> The ground-truth labels for the valence and arousal prediction task were
> obtained from participants self-reporting their experienced levels of valence
> and arousal immediately after being exposed to pictorial stimuli in the
> individual affective baseline task.
>
> > the results don't seem to be so fantastic. They are described as being
> 'better than chance' and I find some appropriate level of discussion of this
> seemingly rather negative result to be lacking.
>
> We agree that further discussion is warranted. We have added all the results
> doing similar things that we can find in the literature and additionally
> explained why a direct comparison is not useful. We wanted to campare fNIRS to
> EEG which entails using sensor groups covering the same parts of the brain. And
> we did get a clear result, namely that fNIRS is substantially better on this
> task (given the particular brain areas). We did not know what to expect, and we
> learned something. Hence this was a successful experiment, which also
> demonstrates something you cannot do with any other dataset.
>
> > The last paragraph in Section 7 seems to suggest that the authors do not have
> an 'understanding (of) what is going on'. This is hardly encouraging.
>
> We have improved the exposition of the results of the second experiment.
> However, it is true that we do not have a tidy theory about what is going on. In
> the first condition (baseline), EEG behaved somewhat as expected, but fNIRS did
> the opposite. For the second condition, we did not necessarily expect a clear
> result (participants were just learning the game) and we did not get much of a
> pattern --- so arguably consistent but boring. The EEG in the third case behaved
> differently from what we expected (we expected more consistenty with the first
> case).
>
> There are many possibilities for what might be going on, and we are keen to
> study this further.
>
> As a dataset paper, the key issue should be whether the data pushes the
> science.  Both experiments are not feasible with existing data sets. Figuring
> out what is going on may also push on additional unique features of the data
> set such as considering what is happening in the game and/or other modalities
> and/or being more careful with the fact that there are three players (we simply
> averaged all three).

---

> > ### Comment · Reviewer_69jK · 2023-08-30
> > **Rebuttal response**
> >
> > Thank you for addressing my comments. I remain uneasy with the uncertainty regarding 'what is going on'. I can't help wonder whether the results and trends observed are impacted by shortcomings in the experimental design or protocol, for example the reliability of self-reported ground-truth labels for the valence and arousal prediction task (even if only a part of the study), or other issues. I appreciate that the experiments reported are not feasible with existing datasets but, if the consistency or reliability of the database and labeling is questionable, then it is also questionable whether or not the database can indeed help push the science. I do not want to suggest that the database and labeling is inconsistent or unreliable, since I cannot tell. Still, the response to this issue isn't encouraging.

---

> > > ### Author Response · Authors · 2023-08-31
> > >
> > > # Response #2 to Reviewer 69jK
> > >
> > > Thank you for your response! We agree that it would be more satisfying if we had
> > > a tidy explanation for why fNIRS performs better than EEG for predicting valence
> > > and arousal, or for why there are a larger number of fNIRS and EEG channels with
> > > a negative correlation between brain data linkage and task performance in the
> > > Minecraft Saturn B task when compared to Minecraft Saturn A. However, the
> > > underlying unknown reality is complex, and this dataset provides the opportunity
> > > to engage with that. The quality of the data needs to be considered independent
> > > of whether it yields tidy results, as we do not know whether to expect such
> > > results in any particular case.
> > >
> > > In more detail, we add that the experiments presented in this paper are
> > > exploratory (i.e., meant to recognize novel patterns in the data, which can
> > > potentially lead to the generation of new hypotheses) rather than confirmatory
> > > (i.e., testing existing hypotheses). Both types of experiments are required for
> > > progress in science [1].  Exploratory experiments are especially appropriate
> > > when the topic of research---in our case, machine learning based on brain
> > > data---is relatively less well-studied compared to other modalities (due to the
> > > inherently challenging nature of collecting brain data) such as images, text, and
> > > audio.
> > >
> > > Regarding the consistency and reliability of the database and labeling---we
> > > understand your uneasiness, but would point out that your comment about results
> > > being impacted by shortcomings in the experimental design or protocol applies
> > > to any experimental study, not just ours. We can, however, attempt to assuage
> > > your concerns about the reliability of self-reported ground-truth labels for
> > > the valence and arousal prediction task.
> > >
> > > As we noted in our [response to
> > > reviewer EAAF](https://openreview.net/forum?id=ZJWQfgXQb6&noteId=sYgFcDDmr4),
> > > the degree to which self-reports are valid measures of emotion varies by the
> > > type of self-report [2]. Notably, self-reports of current emotional experiences
> > > are more likely to be valid than self-reports of emotional experiences in the
> > > past [3]. Given that the self-reported valence and arousal scores used in our
> > > emotion prediction task were collected from participants immediately following
> > > the corresponding pictorial stimulus in the individual affective task, we
> > > believe that it is likely to be a reliable indicator of the participants'
> > > affective state. Finally, we would like to point out that self-reported affect is
> > > a fairly standard procedure in the existing literature on affect prediction from brain data [4-10].
> > >
> > > ## References
> > >
> > > [1] Nosek BA, Ebersole CR, DeHaven AC, Mellor DT. The preregistration
> > > revolution. Proc Natl Acad Sci U S A. 2018 Mar 13;115(11):2600-2606. doi:
> > > 10.1073/pnas.1708274114. PMID: 29531091; PMCID: PMC5856500.
> > >
> > > [2] Mauss, & Robinson, M. D. (2009). Measures of emotion: A review. Cognition
> > > and Emotion, 23(2), 209–237. URL: https://doi.org/10.1080/02699930802204677
> > >
> > > [3] Robinson, M. D., & Clore, G. L. (2002). Episodic and semantic knowledge in
> > > emotional self-report: Evidence for two judgment processes. Journal of
> > > Personality & Social Psychology, 83(1), 198215.
> > >
> > > [4] Soheil Rayatdoost, Yufeng Yin, David Rudrauf, and Mohammad Soleymani. Subject-Invariant Eeg Rep- resentation Learning For Emotion Recognition. In: ICASSP 2021 - 2021 IEEE International Conference on Acoustics, Speech and Signal Processing (ICASSP). 2021, pp. 3955–3959.
> > >
> > > [5] Mohammad Soleymani, Jeroen Lichtenauer, Thierry Pun, and Maja Pantic. A Multimodal Database for Affect Recognition and Implicit Tagging. In: IEEE Transactions on Affective Computing 3.1 (2012), pp. 42–55.
> > >
> > > [6] Soheil Rayatdoost, David Rudrauf, and Mohammad Soleymani. Expression-Guided EEG Representation Learning for Emotion Recognition. In: ICASSP 2020 - 2020 IEEE International Conference on Acoustics, Speech and Signal Processing (ICASSP). 2020, pp. 3222–3226.
> > >
> > > [7] F. Galvão, S. M. Alarcão, and M. J. Fonseca. Predicting Exact Valence and Arousal Values from EEG. In: Sensors (Basel, Switzerland) 21.10 (2021), p. 3414.
> > >
> > > [8] Sander Koelstra, Christian Muhl, Mohammad Soleymani, Jong-Seok Lee, Ashkan Yazdani, Touradj Ebrahimi, Thierry Pun, Anton Nijholt, and Ioannis Patras. Deap: A database for emotion analysis; using physiological signals. In: IEEE transactions on affective computing 3.1 (2011), pp. 18–31.
> > >
> > > [9] Juan Abdon Miranda-Correa, Mojtaba Khomami Abadi, Nicu Sebe, and Ioannis Patras. Amigos: A dataset for affect, personality and mood research on individuals and groups. In: IEEE Transactions on Affective Computing 12.2 (2018), pp. 479–493.
> > >
> > > [10] Stamos Katsigiannis and Naeem Ramzan. DREAMER: A database for emotion recognition through EEG and ECG signals from wireless low-cost off-the-shelf devices. In: IEEE journal of biomedical and health informatics 22.1 (2017), pp. 98–107.

---

> > > > ### Author Response · Authors · 2023-08-31
> > > > **Addendum to response #2 to reviewer 69jK**
> > > >
> > > > # Addendum to response #2 to reviewer 69jK
> > > >
> > > > A quick addendum---we would also like to note that we have run additional experiments, and our results on predicting valence and arousal from fNIRS data are much stronger. We now achieve a classification accuracy of $\approx$ 88%, which is 4.4 times higher than chance. See [this global response](https://openreview.net/forum?id=ZJWQfgXQb6&noteId=l2H2VmcMjA) for details.

---

### Official Review · Reviewer_W3oi · 2023-07-20
**A collaborative rescue dataset simulated in  Minecraft with hybrid signals**

**Rating:** 6
**Confidence:** 1
**Correctness:** Unable to judge.
**Clarity:** Unable to judge.

**Strengths:**

Unable to judge.

**Additional Feedback:**

NA

**Documentation:**

Documents seem to be included all in supplementary_material.pdf

**Ethics:**

Unable to fully judge. But based on the provided documents, the recruitment and data collection process were rigorous and procedurally sound.

**Limitations:**

This work is somehow overwrapped to some big but vague concepts like "rescue" and "human-AI collaboration", which are not accurate in my perspective. The core significance of this work might be physiological signals of participants palying games, which could be complementary to the listerature.

**Opportunities For Improvement:**

- [Motivation] This work does not illustrate the motivation clearly. Why choosing rescue task as the target of collaboration? Does it make sense that participants are playing Minecraft but are required to "imagine" they are doing real rescue? Specifically, in real rescue cases or any other activities, the physiological signals can be strongly affected by real motion instead of sitting in front of a computer.

- [Composition] This work's writing is a bit hard to follow. Besides the area gap, one drawback might be there is no figure in this submission.

**Relation To Prior Work:**

This work discussed on physiological signal dataset as well as works on Minecraft, which seems complete in the reviewer's view.

**Summary And Contributions:**

This submission claim to propose a collaborative rescue dataset, which may include Minecraft player actions and hybrid physiological signals. The submission claim this dataset is beneficial to researches on human-human collaboration even human-AI collaboration.

Unfortunately, since the reviewer is not in the area of social science or neural science, the reviewer is not fully aware of the claimed benefit. The concerns of the reviewer are mainly concentrated in the unclear motivation and the composition of this work.

---

> ### Author Response · Authors · 2023-08-22
> **Response to reviewer W3oi**
>
> Thank you for your feedback! Please see below for responses to your questions
> and concerns.
>
> > [Motivation] This work does not illustrate the motivation clearly. Why
> choosing rescue task as the target of collaboration?
>
> The search and rescue task was chosen due to being time-constrained and
> cognitively, socially, and emotionally demanding. This is coupled with the
> potential for humans and robots to perform complementary team roles in
> real-world USAR scenarios. Please see lines 63--65 of the revised manuscript
> for citations to the relevant literature.
>
> > Does it make sense that participants are playing Minecraft but are required
> to "imagine" they are doing real rescue?
>
> The use of Minecraft-based USAR STEs to study human-machine teaming is
> relatively well-established---they have been used for small-scale studies as
> well as large-scale datasets with numerous associated analyses. Please
> see lines 66--68 of the revised manuscript for citations to the relevant
> literature.
>
> While not the same as a real life SAR mission, the setup is very engaging.
> There is no instruction to to imagine being in a real rescue --- it comes
> naturally. While we expect some difference between the game play and real life
> scenarios, the game is realistic enough that we have a proxy for preliminary
> investigation of many questions such as how stress affects coordination and
> team performance, whether reminding players to communicate when they are not
> doing so is helpful, and what kind of brain patterns are associated with
> leadership and follower styles.
>
> > Specifically, in real rescue cases or any other activities, the
> physiological signals can be strongly affected by real motion instead of
> sitting in front of a computer.
>
> We do not expect participants in a real search and rescue mission to wear
> EEG/fNIRS caps or other sensitive physiological sensors. Rather, the purpose of
> collecting the physiological data is to enable the study of additional aspects
> of theory of mind and human-machine teaming that would not be possible without
> the addition of such sensors.
>
> Please see lines 74--91 of the revised manuscript for a description of the
> motivation behind adding these sensors to the existing ASIST Study 3
> experimental task.
>
> We expect that since the synthetic task environment was constructed to preserve
> the salient aspects of a real-world search and rescue mission, the addition of
> our physiological sensors would not necessarily change the cognitive and
> affective processes that we are interested in studying.
>
> > [Composition] This work's writing is a bit hard to follow. Besides the area
> gap, one drawback might be there is no figure in this submission.
>
> We have made a number of updates to the main paper and supplementary material
> in response to reviewer feedback, including adding a figure. We hope these
> updates help illustrate the motivation and make the writing easier to follow.

---

> > ### Comment · Reviewer_W3oi · 2023-08-29
> > **Response to the Authors**
> >
> > Thank you for your effort on improving the quality of this paper. After scaning the revised version I can grasp a clearer claw of this work.
> >
> > I personally think the authors overwarpped this work into some complex and vague concepts like "rescue" and "Human-AI collaboration". I think they don't have to do so. This dataset is one that records multiple biological signals when participants were playing a game (minecraft). Current researches on AI playing game does rarely involve players' physiological values. Therefore this dataset might be a complement to current literature, although the game involved is not a competitive one like Starcraft, which is somehow depressing. I strongly suggest the authors revise the logic of the draft composition.
> >
> > Overall, I would like to improve the score from 5 to 6. However, the content related to biological/social sciences need further check by in-domain peer reviews.

---

> > > ### Author Response · Authors · 2023-08-31
> > > **Response to reviewer W3oi**
> > >
> > > # Response #2 to Reviewer W3oi
> > >
> > > Thank you for your feedback and your willingness to raise your score.
> > >
> > > > I personally think the authors overwarpped this work into some complex and
> > > vague concepts like "rescue" and "Human-AI collaboration".
> > >
> > > We agree that our wording could be more clear. We have updated the manuscript
> > > (lines 30--32) to emphasize that the focus of our work is not specifically on
> > > search-and-rescue (SAR) missions, but rather on the interesting social behaviors
> > > that can be observed in complex collaborative tasks.
> > >
> > > While 'human-AI collaboration' is a less precisely-defined construct. This is
> > > because the literature on human-AI collaboration (also referred to as 'human-AI
> > > teaming', 'human-machine teaming', and 'human-autonomy teaming') is relatively
> > > less mature compared to say, the literature on human-human teaming [1]. A large
> > > reason for this is that the capabilities of AI agents have only recently begun
> > > to reach levels where it makes sense to conceptualize them as 'teammates' rather
> > > than just 'tools'.
> > >
> > > >  the game involved is not a competitive one like Starcraft
> > >
> > > Yes, the focus of this paper is on cooperation and teamwork under stressful
> > > conditions, rather than competition. That said, studying social interactions in
> > > complex competitive contexts is certainly also interesting.  We would like to
> > > note, however, that we *do provide data from a competitive task*, namely the
> > > competitive ping-pong task (see section 4.3), which can help distinguish
> > > patterns of physiological data in cooperative vs. competitive data. The
> > > competitive ping pong task is one of our unique contributions, in that we
> > > carefully designed the behavioral baseline tasks to ground the data from the
> > > more complex tasks and also provide data on a set of simpler (but still
> > > interesting) tasks.
> > >
> > > ## References
> > >
> > > [1]  Lyons JB, Sycara K, Lewis M and Capiola A (2021) Human–Autonomy Teaming:
> > > Definitions, Debates, and Directions. Front. Psychol. 12:589585. doi:
> > > 10.3389/fpsyg.2021.589585

---

### Official Review · Reviewer_EAAF · 2023-07-21
**A valuable dataset for Theory of Mind-based research but need more experimental results**

**Rating:** 5
**Confidence:** 3
**Correctness:** Suggestions on the experiment section…
**Clarity:** Yes.

**Strengths:**

1. The data collection process and experimental setting are very well-documented;
2. The experimental designs (choice of baseline tasks and specific modalities) are well justified;
3. The initial analysis shows some interesting results, demonstrating the potential utility of the dataset.

**Additional Feedback:**

Suggestions for improvement can be found in the Limitation section.

**Documentation:**

Yes.

**Ethics:**

No.

**Limitations:**

More experiments should be added to Section 7:
1. One suggested domain is affective computing, which is relatively well-studied for multimodal learning. The author should at least consider adding multimodal baselines for some of the proposed baseline tasks;
2. The tasks in the experiment section should be relevant to predicting human beliefs, desires, or mental states;
3. More tasks/experiments that use available modalities other than the brain data and discussion on the results should be included.

**Opportunities For Improvement:**

Major weaknesses:
1. The experiments section is very limited in the following aspects:
i. The dataset is collected for the Theory of Mind-based research, but the initial analysis only explores the tasks of predicting performance score, which is not directly related to predicting human beliefs, desires, or mental states;
ii. The dataset contains an enriched set of modalities, including some most commonly used modalities in multimodal learning (image, text, audio), but the paper only includes unimodal experiments and results and only for the brain data modality;
iii. The paper introduces several baseline tasks, but no corresponding baseline models or preliminary results are found.
2. Only self-report emotion labels are available for the emotion prediction task. Self-reports contain a lot of subjective bias. However, the predictions made by a model trained on the image, audio, transcriptions, or any physiological signals are objective. For example, the model predicts from an observer's perspective by observing the change in participants' facial expressions. How should this perspective difference be mitigated? This also applies to human attributes such as beliefs and desires in Theory of Mind-based prediction.

Minor weaknesses:
1. The descriptions and justifications for tasks/modalities choice should be concise. The discussion takes only 2/9 of the paper, which is much shorter than expected.

**Relation To Prior Work:**

Yes.

**Summary And Contributions:**

The paper presented a large-scale dataset for the Theory of Mind-based Cognitive Architecture for Teams (ToMCAT) project. The dataset contains brain data, physiological signals, images, audio, transcriptions, participants' demographic data, game data featuring the teamwork environment, and self-report questionnaires. The data are collected for several baseline tasks and the simulated search-and-rescue (SAR) missions. The paper presented an initial exploratory analysis of a subset of the modalities. The dataset can serve as a valuable resource for research in machine understanding of human teamwork and facilitate human-machine interaction.

---

> ### Author Response · Authors · 2023-08-22
> **Response to reviewer EAAF (Part 1)**
>
> Thank you for taking the time to review our paper and provide valuable
> feedback! We are glad that you found the dataset well documented, our
> experimental design well justified, and results of our initial analysis
> interesting. Responses to your concerns are provided below.
>
> # Theory of mind, mental states, affective computing.
>
> > The dataset is collected for the Theory of Mind-based research, but the
> initial analysis only explores the tasks of predicting performance score, which
> is not directly related to predicting human beliefs, desires, or mental states;
>
> > The tasks in the experiment section should be relevant to predicting human
> beliefs, desires, or mental states;
>
> > One suggested domain is affective computing, which is relatively well-studied
> for multimodal learning. The author should at least consider adding multimodal
> baselines for some of the proposed baseline tasks;
>
> > More tasks/experiments that use available modalities other than the brain
> data and discussion on the results should be included.
>
> We would like to first point out that we provide not just one, but two initial
> analyses. The first analysis is on the prediction of affect---as reflected by
> self-reported valence and arousal scores on a behavioral task---from brain
> signals. We contend that mental states comprise both cognitive and affective
> components, with the latter playing a large role in human social interactions
> and decision making. Thus, the affect prediction task is related to theory of
> mind, and certainly within the domain of affective computing. We concede that
> our affect prediction benchmark analysis is unimodal rather than
> multimodal---however, as explained later in this response, we are planning on
> performing multimodal analyses on the dataset in the future.
>
> As for the second analysis, on linking team scores with correlated brain
> signals, we agree that this analysis is not directly related to predicting
> human beliefs, desires, and mental states. However, the motivation behind our
> dataset is not only theory of mind-based research, but effective human-machine
> teaming in general. While our hypothesis is that imbuing artificial agents with
> 'machine theory of mind' will make them more effective teammates, we are also
> interested in studying teaming in a broader sense, including understanding
> whether inter-brain synchronization is predictive of team performance.
>
> # Unimodal experiments
>
> > The dataset contains an enriched set of modalities, including some most
> commonly used modalities in multimodal learning (image, text, audio), but the
> paper only includes unimodal experiments and results and only for the brain
> data modality.
>
> We agree that the paper would be improved with additional multimodal
> experiments. However, we believe that our dataset is a sufficiently strong
> contribution even without such experiments. Please see the global rebuttal for
> a more detailed explanation.
>
> # Terminology: Baseline tasks vs. baseline models
>
> > The paper introduces several baseline tasks, but no corresponding baseline
> models or preliminary results are found.
>
> We fear that there is likely a misunderstanding here. We use the term 'baseline
> task' in our paper to refer to behavioral tasks performed by humans. The term
> is completely unrelated to the term 'baseline models' that is commonly found in
> the machine learning literature. For example, in this context, it does not make
> sense to talk about a 'baseline model' for the rest state task, which is simply
> having a participant sit quietly without engaging in any activity.
>
> Additionally, perhaps this is not what you are referring to (and if so, our
> apologies for the mischaracterization), but we did provide preliminary results
> in the 'Exploratory experiments' section of the paper.
>
> # Self-report and bias
>
> > Only self-report emotion labels are available for the emotion prediction task.
> Self-reports contain a lot of subjective bias.
>
> As noted by Mauss and Robinson (2009), the degree to which self-reports are
> valid measures of emotion varies by the type of self-report. Notably,
> self-reports of current emotional experiences are more likely to be valid than
> self-reports of emotional experiences in the past (Robinson & Clore,
> 2002). Given that the self-reported valence and arousal scores used in our emotion
> prediction task were collected from participants immediately following the
> corresponding pictorial stimulus in the individual affective task, we believe
> that it is likely to be a reliable indicator of the participants' affective
> state.

---

> > ### Author Response · Authors · 2023-08-22
> > **Response to reviewer EAAF (Part 2)**
> >
> > > However, the predictions made by a model trained on the image, audio,
> > transcriptions, or any physiological signals are objective. For example, the
> > model predicts from an observer's perspective by observing the change in
> > participants' facial expressions. How should this perspective difference be
> > mitigated? This also applies to human attributes such as beliefs and desires in
> > Theory of Mind-based prediction.
> >
> > The way we see it, the emotion labels ultimately have to be provided by humans,
> > whether from participants via self-report, or external annotators that label
> > the relevant images, transcriptions, or segments of audio or physiological
> > signals. It is not clear that an external annotator will necessarily be able to
> > provide a more accurate label for a participant's affective state than the
> > participant themselves (especially if the self-report is done immediately
> > following the stimulus, as in our task).
> >
> > # Overly-long descriptions and justifications for tasks/modalities
> >
> > > The descriptions and justifications for tasks/modalities choice should be
> > concise. The discussion takes only 2/9 of the paper, which is much shorter than
> > expected.
> >
> > Thank you for the suggestion! We have shortened the descriptions and
> > justifications for the tasks/modalities and expanded the discussion of the
> > results. Additionally, we have expanded the related work section and added a
> > conclusion section.
> >
> > # References
> >
> > Mauss, & Robinson, M. D. (2009). Measures of emotion: A review. Cognition and
> > Emotion, 23(2), 209–237. URL: https://doi.org/10.1080/02699930802204677
> >
> > Robinson, M. D., & Clore, G. L. (2002). Episodic and semantic knowledge in
> > emotional self- report: Evidence for two judgment processes. Journal of
> > Personality & Social Psychology, 83(1), 198215.

---

> > > ### Comment · Reviewer_EAAF · 2023-08-28
> > >
> > > Dear authors,
> > >
> > > Firstly, thank you for your detailed clarifications and rebuttals. The latest revision of the paper has been greatly improved. The reviewer agrees with the author's point that the brain data is much less studied compared to the other ones (image, audio, text) and is of great interest to the research community. However, since the paper is presented as a multimodal benchmark (especially since it is labeled with the keyword multimodal), the reviewer believes a minimum of multimodal experiments/explorations should be included, which will serve as a valuable starting point and reference for future use of the benchmark by the community. Given the reason above, the reviewer decided to raise the rating to 5.

---

> > > > ### Author Response · Authors · 2023-08-28
> > > > **Response to Reviewer EAAF**
> > > >
> > > > We are pleased that you found our revised manuscript greatly improved!
> > > >
> > > > We would like to clarify that the keyword 'multimodal' was chosen to describe the dataset itself, rather than the benchmark analyses. While we still believe that the paper is a strong enough contribution for this track without the multimodal benchmark analyses, we respect your assessment and appreciate your increasing our score!

---

### Official Review · Reviewer_A9Aa · 2023-07-21
**The paper introduces an interesting dataset and idea to include AI with humans in a multisensior scenario. The paper is not well structured, not clear and difficult to read.**

**Rating:** 6
**Confidence:** 4

**Strengths:**

The authors are reporting a multi modal dataset to enhance the human performance with AI agents.The idea is interesting for the broader research community. Note that it is not clear whether the data collection have an ethical clearance.

**Additional Feedback:**

Would be interesting to see if they could try to combine the EEG and fNIRS data for example and how this changes the performance of the models.
It is noted that the data were collected in 6 period time. Would be interesting to see if there is any analysis on the results regarding the long duration of acquiring these data. For example, if the results are affected with time and how.

**Clarity:**

Although the paper introduces a very interesting topic, the paper is not clear written.

For example, the authors mentioned that when there were cases when members from the research team had to fill in the places when participants were missing, but they do not state how many times this happened (line 105-107). In line 111: a tutorial mission is mentioned but not the duration of it.

There are inconsistencies in the information given, for example: the number of participants (lines 1-2. lines 510-515), the train-test-validation split, the given compensation/benefit to the participants.


**Correctness:**

The authors report two different exploratory experiments: (i) predicting valence and arousal and (ii) correlation of brain signals with scores on three different tasks. Would be interesting to see results on using the other information as well as there many other modalities in the acquired dataset.

The analysis for the classification results on valence and arousal is limited to reporting only accuracy numbers without any further analysis of these results. The reported confusion matrix is 5x5 although it is binary classification problem.
Furthermore there are some typos in this subsection. To mention a few:
line 322: we see that the accuarcy of the fNIRS baseline ....
line 324: ... the accuracy of the EEG baseline ...
line 324: ... 38.91% for valence ...

caption of figure 3: should be EEG and not fNIRS


**Documentation:**

The clarity of the paper regarding the dataset description and the experimental setup should be improved. As mentioned previously would be great to include:
- a clear description of the dataset
- a clear description of the experimental setup. A figure illustrating the experimental setup would be highly appreciated.

The authors provide a url for downloading the dataset, although the access to the dataset is not easy.

**Ethics:**

It is not clear from the paper whether the dataset has undergo an ethical clearance or not.

**Limitations:**

The limitation section provides enough details. Worth to mention thought that the missing information of all modalities does not help the clarity of the paper.




**Opportunities For Improvement:**

Suggestions for the authors:
- provide a clear description of the dataset
- provide a clear description of the experimental setup. A figure illustrating the experimental setup would be highly appreciated.

In line 29: There are 2 different setups mentioned (purely human and hybrid). Would be interesting to see results on these 2 different setups along with analysis. For example, which one gives better results and why.



**Relation To Prior Work:**

The section of the related work should be re-written to reflect the current state of the art research in this field. At the moment the related work section is focusing on 3 aspects: the ASIST Study 3 dataset, the fNIRS2MW dataset and minecraft related research.

**Summary And Contributions:**

The paper introduces an interesting dataset and idea on how to include AI with humans in a multidimensional scenario, using data from fNIRS, EEG, skin conductance, heart rate, eye tracking, face images, audio with ASR, game screenshots, game data, performance data, demographic data, and questionnaires.  The dataset contains data from 120 humans (40 teams with 3 humans each), although different number is reported in the supplementary file.

---

> ### Author Response · Authors · 2023-08-22
> **Response to reviewer A9Aa (Part 1)**
>
> We are pleased to hear that you think our dataset would be of interest to the
> broader research community, and are grateful for your valuable feedback. We
> have provided responses to your concerns below.
>
> # Strengthening related work section
>
> > The section of the related work should be re-written to reflect the current
> state of the art research in this field.
>
> We have significantly expanded our Related Work section in the revised
> manuscript to address your concerns above. We have also expanded our analysis
> section (section 6 in the revised manuscript) to add references to other
> relevant literature.
>
> # Ethical clearance
>
> > Note that it is not clear whether the data collection have an ethical clearance.
>
> We had already described our ethical review process on lines 250 -- 273 of the
> originally submitted supplementary material. We have now added a pointer to it
> in the revised version of the main paper (line 160).
>
> # Dataset accessibility
> > the access to the dataset is not easy.
>
> We respectfully disagree. We have taken the following steps to make the dataset
> accessible:
> 1. As you have noted, we have provided a public URL to the dataset.
> 2. We provide a Datasette-based web interface for querying the database without
>    requiring the user to download the data (this was noted by reviewer yyXM as
>    a factor that makes the dataset very accessible).
> 3. We use open and widely used formats (SQLite and CSV).
> 4. We provide structured metadata using web standards, as required by NeurIPS.
>
> However, we are of course open to any suggestions you may have on how to
> make the dataset even more accessible than it already is.
>
> # Inconsistent number of participants
>
> > The dataset contains data from 120 humans (40 teams with 3 humans each),
> although different number is reported in the supplementary file.
>
> Both the main paper and the supplementary material report the same number of
> teams (40). Note that the total number of unique participants in the dataset is
> 102 (see
> [here](https://tomcat.ivilab.org/tomcat?sql=select%0D%0A++count%28id%29%0D%0Afrom%0D%0A++participant%0D%0Awhere%0D%0A++id+%3E+0%0D%0A++and+id+%3C+999)), rather than 120, since experimental confederates filled in for certain
> tasks for no-show participants or participants that left in the midst of a
> group session. The total number of experimental confederates represented in the
> dataset is 9 (see
> [here](https://tomcat.ivilab.org/tomcat?sql=select%0D%0A++count%28id%29%0D%0Afrom%0D%0A++participant%0D%0Awhere%0D%0A++id+%3E+999)).
> Note that certain confederates participated in multiple group sessions, which
> is why the sum of number of naive participants (102) and the number of
> confederates (9) does not equal 120.
>
> > There are inconsistencies in the information given, for example: the number
> of participants (lines 1-2. lines 510-515),
>
> In our previous submission, we reported data from 85 subjects for fNIRS and 79
> for EEG. This discrepancy from the expected can attributed to our initial data
> extraction pipeline, which excluded subjects with corrupted Minecraft data. As
> a consequence, we inadvertently overlooked a subset of valid baseline task
> data.
>
> To rectify this, we refined our data extraction methodology. Our updated
> pipeline segregates the physiological data for each task and subsequently
> attempts to label them. As a result of this enhancement, we have successfully
> retrieved fNIRS data from 102 individual affective task sessions and EEG data
> from 99 individual affective task sessions, with the missing EEG data for three
> sessions attributable to us having to send in one of our EEG amplifiers for
> repair. We have updated the relevant numbers in the revised supplementary
> material (lines 562--563).
>
> > the train-test-validation split, the given compensation/benefit to the participants.
>
> We were not able to find the inconsistencies in the train-test-validation split
> and the given compensation/benefit to the participants that you refer to. We
> would appreciate pointers to the lines in the manuscript where you found this
> inconsistency.

---

> > ### Author Response · Authors · 2023-08-22
> > **Response to reviewer A9Aa (Part 2)**
> >
> > > Provide a clear description of the dataset.
> >
> > We had hoped that our 12-page long datasheet in the supplementary material
> > would have served as a sufficiently clear description of the dataset, but
> > evidently we could do better. We would appreciate any concrete suggestions you
> > may have on how to make our description clearer (e.g., if there are specific
> > aspects of the dataset that you found insufficiently or improperly documented).
> >
> > > provide a clear description of the experimental setup. A figure illustrating
> > the experimental setup would be highly appreciated.
> >
> > Following your suggestion (which was also echoed by reviewer W3oi), we have
> > added a figure illustrating the experimental setup to the revised manuscript
> > (Figure 1).
> >
> > > Worth to mention thought that the missing information of all modalities does
> > not help the clarity of the paper.
> >
> > We are not sure we understand which section of the paper you are referring to
> > here. If you are referring to our mention of missing data in the Limitations
> > section in the supplementary material, we respectfully disagree---we believe
> > that being upfront about missing data is extremely important for a dataset
> > paper, and can unlikely to detract from clarity since it is in the
> > supplementary material rather than the main paper itself.
> >
> > > In line 29: There are 2 different setups mentioned (purely human and hybrid).
> > Would be interesting to see results on these 2 different setups along with
> > analysis. For example, which one gives better results and why.
> >
> > We agree completely! This is one of the analyses we will prioritize for the
> > many future publications based on this dataset that we are planning to write.
> > However, we believe that this paper is a significant enough contribution to the
> > literature as it currently is (i.e., without additional experiments comparing
> > purely human and hybrid setups) to warrant publication in this venue.
> >
> > > Would be interesting to see results on using the other information as well as
> > there many other modalities in the acquired dataset.
> >
> > We agree! Please see the global rebuttal for our response to this issue, which
> > was also raised by reviewer EAAF.
> >
> > > The analysis for the classification results on valence and arousal is limited
> > to reporting only accuracy numbers without any further analysis of these
> > results.
> >
> > We have now expanded the valence/arousal classification section with further
> > analysis of the results (lines 344--346 of the revised manuscript).
> >
> > > The reported confusion matrix is 5x5 although it is binary classification
> > problem.
> >
> > We would like to clarify that this is not a binary classification problem. We
> > had participants self-report valence and arousal on a 5-point scale. This was
> > noted in lines 176--178 of the original paper. In the revised manuscript, we
> > now explicitly mention that the scale used is a 5-point scale (line 261).
> >
> > > Although the paper introduces a very interesting topic, the paper is not clear
> > written.
> >
> > We are sorry to hear that you found our paper not clearly written. In response
> > to reviewer feedback, we have significantly expanded our related work section,
> > added a figure and a conclusion section, and added additional discussion of the
> > results of the exploratory experiments. We hope that these changes are
> > sufficient to raise your assessment of the clarity of our paper.
> >
> > > For example, the authors mentioned that when there were cases when members from
> > the research team had to fill in the places when participants were missing, but
> > they do not state how many times this happened (line 105-107).
> >
> > Out of the 1014 task instances for which data was supposed to be recorded for
> > regular participants, 147 (i.e., 14.5%) had experimenters filling in for
> > participants. We have added this information to lines 198--200 of the revised
> > manuscript.
> >
> > > In line 111: a tutorial mission is mentioned but not the duration of it.
> >
> > The tutorial mission is 20 minutes long. We have added this information to line
> > 204 of the revised manuscript. Our apologies for the oversight.

---

### Official Review · Reviewer_yyxM · 2023-07-22
**Large and accessible bio-data dataset from Minecraft games, but needs clearer motivation, clearer relation to prior work, and updated analysis**

**Rating:** 6
**Confidence:** 4
**Clarity:** The writing is clear and easy to follow.

**Strengths:**

The authors present a large collection of data on people playing together, including setups that involve AI. This data can be used for a variety of tasks.

The authors provide a web interface for querying the database, making it very accessible (e.g. without requiring download).

The writing is clear and easy to follow.

**Additional Feedback:**

Details about the COVID-19 health screener are not needed.

****************************
Post-discussion: Thanks to the reviewers for the thorough consideration of reviewer feedback. The authors have addressed many of my concerns, and I find the manuscript significantly improved. As a result I am moving my score up to a 6.

**Correctness:**

The collection seems to have been conducted appropriately.

The methods used in the analyses should be further substantiated (described above). For the regression, additional details are needed, such as significance for each coefficient.

**Documentation:**

The authors provide a database schema on their website, describe the collection mechanism in the paper, and provide a datasheet in the supplementary materials.

A description of the participant pool seems to be missing (general demographics). This is needed to understand representation and potential biases in the data.

**Ethics:**

The team engaged in consent and compensation, provide the consent form and recruitment materials on their website, and provide a datasheet with additional details.

**Limitations:**

The authors mention several limitations in preparing the dataset (unclear alignment between data streams, accounting for a researcher filling in for a participant).

It is unclear what limitations arise from the recruitment method, or the specific pool of participants, as demographics are not shared even in aggregate. It would be beneficial to share recruitment methods and general demographics to enable assessment of limitations in representation.

It would be beneficial for the authors to add a limitations subsection to the paper, explicitly describing such limitations.

**Opportunities For Improvement:**

The paper would benefit from clarifying the problem or need that this dataset is designed to fill. What is the problem with the existing similar datasets? What need does this dataset fill that others do not? What new potential does this dataset unlock?

Relatedly, the connection to Theory of Mind should be strengthened, or removed. Theory of Mind is highlighted in the introduction and dataset name, but is not discussed in the body of the work, so the connection with the collection mechanism and evaluation is unclear.

The paper would benefit from a description of the recruitment process (e.g. where were participants recruited from, and how were they recruited?), and description of the participant pool (e.g. gender distribution, age, etc.).

The baseline analyses would benefit from grounding in prior work. In the current paper presentation, it is unclear why these two analyses were chosen (why predicting valence/arousal from brain scan data, and why predicting task performance from correlation)? It is also unclear how the methods relate to the state-of-the-art in methods for these tasks (is the state-of-the-art to use LSTM and regression, respectively?) and performance (how do the reported numerical results compare to existing work?).

The authors mention several challenges in preparing the dataset, including alignment between asynchronous data streams and accounting for the researcher filling in for a participant who left. However, how these challenges were handled does not seem to be documented and shared.

The authors outline that their dataset is most similar to the ASIST Study 3 dataset, and specify the ways in which their dataset differs. However, they do not explain why these differentials are important or demonstrate experimentally the value of their dataset over such past ones.

The paper would benefit from a conclusion section.

**Relation To Prior Work:**

The related work section is incomplete. I would recommend expanding it into three subsections on prior datasets, prior task designs, and prior work on understanding theory of mind in gameplay. Given that the tasks that participants engaged in are cited as one of the three main contributions in the introduction, the related work section would benefit from a section on prior tasks specifically, as well as expanding the other two subtopics. The evaluation/analysis is also missing grounding/comparison to prior work.

**Summary And Contributions:**

This work presents a database of primarily bio data from teams of 3 people playing 6 phases of a search-and-rescue game similar to Minecraft. The data contains two types of brain scan data, and other bio data (skin, heart rate, spoken data, demographics, etc.). The authors also provide two analyses, one predicting valence and arousal from brain scan data, and a second using regression to predict task performance from correlation between players’ brain scan data.

---

> ### Author Response · Authors · 2023-08-22
> **Response to reviewer yyXM (Part 1)**
>
> Thank you for your valuable feedback and suggestions! We are pleased to hear
> that you found our dataset accessible and our writing clear. We
> respond to some of your comments and feedback below.
>
> # Clarifying the motivation for the dataset
>
> > The paper would benefit from clarifying the problem or need that this dataset
> is designed to fill. What is the problem with the existing similar datasets?
> What need does this dataset fill that others do not?
>
> Existing datasets generally have physiological sensor data for tasks of
> relatively limited cognitive complexity (e.g. the fNIRS2MW and EEGEyeNet
> datasets), or data on complex human-machine teaming tasks (e.g., the ASIST
> Study 3 dataset), but without any physiological sensor data. Ours is the only
> dataset that features both.
>
> Humans are not robots - their mental states include both cognitive
> and affective components, with the latter not been extensively studied
> in the human-machine teaming and hyperscanning literature despite the crucial
> role it plays in human social interactions and decision making.
>
> > What new potential does this dataset unlock?
>
> Adding physio sensors allows us to study affect in teams in a way that is
> grounded in actual physiological data (as opposed to merely self-reported
> affect or coding by external observers). As Mauss and Robinson [16] note,
> there is no 'gold standard' measure of emotion---self-report, physiological,
> and behavioral measures are sensitive to different types of emotional state,
> and we include all three in our experimental design.
>
> With additional layers of annotation, more potential research avenues will be
> unlocked. For example, one of the analyses we plan to perform on this dataset
> in the future is to study multiparty phonetic entrainment during the course of
> participants conducting a collaborative task while communicating via speech.
> This is enabled by us conducting the speech elicitation task during the
> pre-session, which provides baseline data to which we can compare
> participants' speech during the simulated USAR mission to see if their speech
> patterns get more aligned over the course of the task. Another layer of
> annotation could involve labeling participant utterances with sentiment or
> dialog act labels in order to develop models for sentiment and dialog act
> classification in task-related dialog.
>
> # Aggregate demographic data and discussion of representation and biases
>
> > A description of the participant pool seems to be missing (general
> demographics). This is needed to understand representation and potential biases
> in the data.
>
> We have added the following to the revised supplementary material:
> - Aggregate demographic data (lines 194--197).
> - A discussion of representation and biases in the data (lines 80--94).

---

> > ### Author Response · Authors · 2023-08-22
> > **Response to reviewer yyXM (part 2)**
> >
> > # Grounding baseline analyses in prior work, comparison to state of the art
> >
> > > The baseline analyses would benefit from grounding in prior work. In the
> > current paper presentation, it is unclear why these two analyses were chosen
> > (why predicting valence/arousal from brain scan data, and why predicting task
> > performance from correlation)?  It is also unclear how the methods relate to
> > the state-of-the-art in methods for these tasks (is the state-of-the-art to use
> > LSTM and regression, respectively?) and performance (how do the reported
> > numerical results compare to existing work?).
> >
> > ## Predicting valence/arousal from brain scan data
> >
> > ### Motivation
> >
> > 1. Brain Activation Related to Emotional Processing:
> >     * fNIRS measures changes in oxygenated and deoxygenated hemoglobin, which can
> >     give you insights into which areas of the brain are being activated. For
> >     emotional processing, you might expect activity in regions such as the
> >     prefrontal cortex (PFC), which plays a role in emotional regulation [1].
> >     * EEG provides information about the electrical activity in the brain. Certain
> >     EEG patterns and event-related potentials (ERPs) are associated with
> >     emotional processing. For example, the late positive potential (LPP) is known
> >     to be modulated by emotional valence and arousal [2].
> >
> > 2. Temporal Dynamics of Emotional Responses:
> >     * Since EEG has high temporal resolution, you can observe the
> >     millisecond-by-millisecond evolution of the brain's response to each image,
> >     capturing rapid emotional reactions [3].
> >     * fNIRS, while not as temporally precise as EEG, can still provide insights
> >     into the slower hemodynamic responses associated with prolonged or sustained
> >     emotional processing.
> >
> > 3. Cognitive Aspects of Emotional Processing:
> >     * Certain EEG frequencies (like theta and gamma) can be linked to cognitive
> >     aspects of emotion, such as attention, memory, and perception [4].
> >     * Buhle et. al. [5] demonstrates using fMRI how cognitive regulation
> >     techniques, such as reappraisal, engage specific brain areas, including the
> >     dorsolateral prefrontal cortex dlPFC, to modulate emotional responses . Since
> >     fNIRS operates on principles similar to fMRI, it can be posited that fNIRS
> >     might also suggest cognitive regulation or control over emotional responses.
> >
> > ### Comparison to state of the art
> >
> > In lines 309--339 of the revised paper, we have added references to other
> > works in the literature that attempt tasks similar to this one, along
> > with summaries of their results. We also discuss the limitated usefulness of
> > comparing our results with theirs.
> >
> > ## Linking temporal correlation of brain signals with scores
> >
> > ### Motivation
> >
> > We have expanded the paper with additional motivation for this experiment
> > (lines 348--364 of the revised manuscript).
> >
> > ### Comparison to state of the art
> >
> > 1. Unique dataset: Our methods are unique in that we examine the brain
> >    holistically with mulimodal recordings in Ping Pong Cooperative Task
> >    (laboratory settings) and Minecraft missions (naturalistic settings) to
> >    capture the complex nature of shared mental models and better understand the
> >    multifaceted processes of collaboration.
> > 2. Similar analysis methods as existing studies: We are not introducing any
> >    novel analysis techniques with this experiment. Instead, we are employing
> >    methods congruent with existing studies [9, 11]. Specifically, we are
> >    determining the similarity or dependency among brain signals across subjects
> >    and examining how this connection correlates with team performance.
> >
> > ## Significance for each linear regression coefficient
> >
> > > For the regression, additional details are needed, such as significance for
> > each coefficient.
> >
> > We have added significance in the form of p-values for each coefficient to
> > Table 2.
> >
> > ## Aligning asynchronous data streams and accounting for experimenter substitutions.
> >
> > > The authors mention several challenges in preparing the dataset, including
> > alignment between asynchronous data streams and accounting for the researcher
> > filling in for a participant who left. However, how these challenges were
> > handled does not seem to be documented and shared.
> >
> > ### Aligning asynchronous data streams
> >
> > We had already described the procedure for alignment between asynchronous data streams in
> > the supplementary material (Section I.1, lines 779--821).
> > In the revised supplementary material, the procedure is in Section
> > J.1 (lines 799--841). Please let us know if you would like more details!

---

> > > ### Author Response · Authors · 2023-08-22
> > > **Response to reviewer yyXM (Part 3)**
> > >
> > > ### Accounting for researchers filling for a participant
> > >
> > > The `participant` column in the [`data_validity`
> > > table](https://tomcat.ivilab.org/tomcat/data_validity) in the ToMCAT database
> > > explicitly keeps track of who participated in each task (synonymous with the word 'phase'
> > > used earlier in this response). This makes it possible for researchers to
> > > either ignore this data if they so choose (which may not be necessary,
> > > depending on the research question being explored).
> > > Note that only about 15% of the task data (see
> > > [here](https://tomcat.ivilab.org/tomcat?sql=with+experimenters%28n%29+as+%28%0D%0A++select%0D%0A++++count%28*%29%0D%0A++from%0D%0A++++data_validity%0D%0A++where%0D%0A++++participant+%3E+999%0D%0A++++and+modality+%3D+%27eeg%27%0D%0A%29%2C%0D%0Aall_participants%28n%29+as+%28%0D%0A++select%0D%0A++++count%28*%29%0D%0A++from%0D%0A++++data_validity%0D%0A++where%0D%0A++++participant+%3E+0%0D%0A++++and+modality+%3D+%27eeg%27%0D%0A%29%0D%0Aselect%0D%0A++experimenters.n*100.0%2F%0D%0A++all_participants.n%0D%0Afrom%0D%0A++experimenters%2C%0D%0A++all_participants)
> > > have an experimenter substituting for a real participant. Even if one chooses
> > > to completely ignore this data (which may not even be necessary, depending on
> > > the research question being addressed), there would still remain have a large
> > > amount of rich, multimodal data to work with.
> > >
> > > # Expanded Related Work section.
> > >
> > > > Relatedly, the connection to Theory of Mind should be strengthened, or
> > > removed. Theory of Mind is highlighted in the introduction and dataset name,
> > > but is not discussed in the body of the work, so the connection with the
> > > collection mechanism and evaluation is unclear.
> > >
> > > > Explain why the differences between the ASIST Study 3 dataset and the ToMCAT
> > >   dataset matter, and demonstrate experimentally the value of their dataset
> > >   over the past ones.
> > >
> > > > Related work section is incomplete - expand into three subsections on prior
> > > datasets, prior task designs, prior work on understanding ToM in gameplay.
> > >
> > > We have significantly expanded our Related work section in the revised
> > > manuscript to address your three concerns above. While we did not follow the
> > > exact subsectioning scheme you suggested (we felt that a different structure
> > > made a bit more sense for this paper), we hope that it is sufficiently clear
> > > now.
> > >
> > > # Recruitment process, limitations section
> > >
> > > > The paper would benefit from a description of the recruitment process (e.g.
> > > where were participants recruited from, and how were they recruited?)
> > >
> > > We had already provided a description of the recruitment process in lines
> > > 69--80 of the supplementary material. We have now added a pointer to these
> > > details in the revised version of the main paper (line 160), and moved them to a
> > > separate section in the revised supplementary material (Appendix B)
> > > to make them easier to find.
> > >
> > > > It would be beneficial for the authors to add a limitations subsection to the
> > > paper, explicitly describing such limitations.
> > >
> > > We had already included a limitations section in the supplementary material
> > > (Appendix J, lines 848--861), and included a pointer to it in the Checklist at
> > > the end of the paper. In the revised supplementary material, the limitations
> > > section is in Appendix L. We felt that it was appropriate to include the
> > > limitations in the supplementary material, following the example of a previous
> > > best paper award winner in the NeurIPS Datasets and Benchmarks Track [15].
> > >
> > > > Add a conclusion section.
> > >
> > > We have added a conclusion section in the revised manuscript.
> > >
> > > > playing 6 phases of a search-and-rescue game
> > >
> > > We would like to clarify that while there are six phases in each 'group
> > > session', only the last three of the phases are conducted in Minecraft.
> > > missions. The first three phases are behavioral tasks (rest state, affective
> > > task, ping-pong task), the fourth phase is a 'tutorial' Minecraft mission, and
> > > the last two phases are the actual Minecraft search-and-rescue missions.

---

> > > > ### Author Response · Authors · 2023-08-22
> > > > **Response to reviewer yyXM (Part 4)**
> > > >
> > > > # References
> > > >
> > > > - [1] Ehlis AC, Schneider S, Dresler T, Fallgatter AJ. Application of functional near-infrared spectroscopy in psychiatry. Neuroimage. 2014 Jan 15;85 Pt 1:478-88. [Link](https://doi.org/10.1016/j.neuroimage.2013.03.067)
> > > >
> > > > - [2] Olofsson JK, Nordin S, Sequeira H, Polich J. Affective picture processing: an integrative review of ERP findings. Biol Psychol. 2008 Mar;77(3):247-65. [Link](https://doi.org/10.1016/j.biopsycho.2007.11.006)
> > > >
> > > > - [3] Hajcak G, MacNamara A, Olvet DM. Event-related potentials, emotion, and emotion regulation: an integrative review. Dev Neuropsychol. 2010;35(2):129-55. [Link](https://doi.org/10.1080/87565640903526504)
> > > >
> > > > - [4] Cavanagh JF, Shackman AJ. Frontal midline theta reflects anxiety and cognitive control: meta-analytic evidence. J Physiol Paris. 2015 Feb-Jun;109(1-3):3-15. [Link](https://doi.org/10.1016/j.jphysparis.2014.04.003)
> > > >
> > > > - [5] Buhle JT, Silvers JA, Wager TD, Lopez R, Onyemekwu C, Kober H, Weber J, Ochsner KN. Cognitive reappraisal of emotion: a meta-analysis of human neuroimaging studies. Cereb Cortex. 2014 Nov;24(11):2981-90. [Link](https://doi.org/10.1093/cercor/bht154)
> > > > - [6] D. McEwan, G. R. Ruissen, M. A. Eys, B. D. Zumbo, and M. R. Beauchamp, “The Effectiveness of Teamwork Training on Teamwork Behaviors and Team Performance: A Systematic Review and Meta-Analysis of Controlled Interventions,” PLoS ONE, vol. 12, no. 1, p. e0169604, Jan. 2017, doi: 10.1371/journal.pone.0169604.
> > > >
> > > > - [7] J. A. Cannon-Bowers and E. Salas, “Reflections on shared cognition,” J. Organiz. Behav., vol. 22, no. 2, pp. 195–202, Mar. 2001, doi: 10.1002/job.82.
> > > >
> > > > - [8] D. McEwan, G. R. Ruissen, M. A. Eys, B. D. Zumbo, and M. R. Beauchamp, “The Effectiveness of Teamwork Training on Teamwork Behaviors and Team Performance: A Systematic Review and Meta-Analysis of Controlled Interventions,” PLoS ONE, vol. 12, no. 1, p. e0169604, Jan. 2017, doi: 10.1371/journal.pone.0169604.
> > > >
> > > > - [9] G. Fronda and M. Balconi, “What hyperscanning and brain connectivity for hemodynamic (fNIRS), electrophysiological (EEG) and behavioral measures can tell us about prosocial behavior.,” Psychology & Neuroscience, vol. 15, no. 2, pp. 147–162, Jun. 2022, doi: 10.1037/pne0000260.
> > > >
> > > > - [10] P. Reddish, R. Fischer, and J. Bulbulia, “Let’s Dance Together: Synchrony, Shared Intentionality and Cooperation,” PLoS ONE, vol. 8, no. 8, p. e71182, Aug. 2013, doi: 10.1371/journal.pone.0071182.
> > > >
> > > > - [11] T. Liu, L. Duan, R. Dai, M. Pelowski, and C. Zhu, “Team-work, Team-brain: Exploring synchrony and team interdependence in a nine-person drumming task via multiparticipant hyperscanning and inter-brain network topology with fNIRS,” NeuroImage, vol. 237, p. 118147, Aug. 2021, doi: 10.1016/j.neuroimage.2021.118147.
> > > >
> > > > - [12] L. Astolfi et al., “Imaging the social brain by simultaneous hyperscanning during subject interaction,” IEEE Intelligent Systems, vol. 26, no. 5, pp. 38–45, 2011, doi: 10.1109/MIS.2011.61.
> > > >
> > > > - [13] N. Kaiser and E. Butler, “Introducing Social Breathing: A Model of Engaging in Relational Systems,” Front. Psychol., vol. 12, p. 571298, Apr. 2021, doi: 10.3389/fpsyg.2021.571298.
> > > >
> > > > - [14] N. J. Bourguignon, S. L. Bue, C. Guerrero-Mosquera, and G. Borragán, “Bimodal EEG-fNIRS in Neuroergonomics. Current Evidence and Prospects for Future Research,” Front. Neuroergonomics, vol. 3, p. 934234, Aug. 2022, doi: 10.3389/fnrgo.2022.934234.
> > > >
> > > > - [15] Fan et al. [MineDojo: Building Open-Ended Embodied Agents with Internet-Scale Knowledge](https://openreview.net/forum?id=rc8o_j8I8PX). In: NeurIPS. 2022.
> > > >
> > > > - [16] Mauss, & Robinson, M. D. (2009). Measures of emotion: A review.
> > > >   Cognition and Emotion, 23(2), 209–237. URL:
> > > >   https://doi.org/10.1080/02699930802204677

---

### Author Response · Authors · 2023-08-22
**Global response**

We would like to thank all the reviewers for their valuable feedback.  We were
pleased to hear that reviewers found our dataset accessible (yyXM), interesting
(A9Aa, EAAF), and well-documented (EAAF), our experimental design
well-justified (EAAF), recruitment and data collection process rigorous and
procedurally sound (W3oi), and our writing clear (yyXM).

At the same time, we acknowledge that there was room for improvement in the
paper. Below, we list issues raised by more than one reviewer.

# Connection to existing literature

Multiple reviewers (yyXM, A9Aa) noted that the paper would benefit from
strengthening the connections to existing literature and datasets. We have
significantly expanded our Related Work section in the revised manuscript to
address your concerns above. We have also expanded our analysis section
(section 6 in the revised manuscript) to add references to other relevant
literature. We also now explicitly point out our dataset is the largest
open-access fNIRS dataset by a factor of 13.5.

# Unimodal experiments

Reviewers A9Aa and EAAF noted our lack of multimodal experiments in this paper
as an opportunity for improvement.

We are eager to perform more multimodal experiments in the future, and we
expect additional insights from combining modalities. However, we believe that
our dataset is sufficiently large, rich, and well-motivated that the broader
research community will benefit more from us sharing the dataset in a timely
manner, allowing researchers to explore the data and devise their own
experiments (even if they decide to go with unimodal experiments, which we
think unlikely), rather than us keeping the data to ourselves while we wrangle
this highly complex data. In this way, we follow the example of the ASIST Study
2 and ASIST Study 3 datasets, which are comparable to ours in terms of
complexity and size. Additionally, we would like to note that datasets with
unimodal experiments have been well-received and accepted to this track in the
past (e.g., the fNIRS2MW dataset). We focused on using brain data in this paper
due to our interest in inter-brain synchrony, and because brain data
(especially fNIRS) is relatively less studied in the machine literature
compared to the image, text, and audio modalities.

# Lack of figure

Reviewers A9Aa and W3oi noted that a figure illustrating the experimental
setup would be helpful. We have added one in the revised manuscript
(Figure 1).

# Potential for improvement in the signal filtering and synchronization process

We wish to note that due to the complexity of our dataset dataset, our approach
to signal filtering and synchronization is continuously being refined, and we
expect our results to evolve to become more accurate and yield deeper
insights.

To reflect our improved understanding of the data collected and changes to the
filtering and synchronization process between the original submission and the
revised version, we resampled the signal to the desired sampling rate and
interpolated the signals to synchronize them to a unified time series. We
modified the supplemental material to reflect this change in section J.

# Reanalysis of data for preliminary studies

In our previous submission, we reported data from 85 subjects for fNIRS and 79
for EEG. This discrepancy from the expected was attributed to our initial data
extraction pipeline, which excluded subjects with corrupted Minecraft data. As
a consequence, we inadvertently overlooked the processing of the baseline task
data. Furthermore, we improved our signal filtering and synchronization
pipeline which enhanced the quality of the data.

With the enhancement of the data, we conducted a comprehensive reanalysis to
obtain the most recent results, which accurately represent the present
condition of our dataset. The new results are reflected in Table 1 and Table 2
of the main paper. Note that overall results for valence and arousal prediction
are not significantly impacted. For the correlation experiments, we see that
the channels that act as the strongest predictors of performance do change, but
since we did not start with a hypothesis about which channels would be the best
predictors (rather, our experiment is very much exploratory in nature), the
qualitative interpretation of results provided in the discussion section
remains unchanged.

# Other improvements

In addition to addressing concerns explicitly raised by reviewers, we have also
made a number of other improvements to the paper.
- We have increased the font size in Figures 2 and 3 in the supplementary
  material for increased legibility.
- We have fixed a few errant typos in the paper, and improved the formatting
  for some portions.
- We have added additional details about our EEG electrode setup in the
  supplementary material.

We thank the reviewers again for taking the time to carefully review our paper,
and hope we have adequately addressed their concerns with our revised
manuscript and author responses.

---

### Author Response · Authors · 2023-08-28
**Update (2023-08-28)**

We have uploaded a new version of the manuscript with the following updates:

1. We fixed a minor bug in our synchronization code that had shifted the signal
   timestamps by a small amount. We also discovered and fixed timestamp issues for EEG signals of experiment exp_2022_10_18_10, and we removed EEG signals from experiment exp_2022_10_24_12 in our analysis to address the timestamp issues. This has resulted in changes in the slopes and
   significance values in Table 2---notably:
   - For the Ping Pong Cooperative task, fNIRS channels now tend to display
     more positive slopes and EEG channels tend to display more negative
     slopes, rather than the other way around.
   - For the Minecraft Saturn B mission, there is a larger number of fNIRS
     channels with negative correlation between brain data linkage and task
     performance when compared to Minecraft Saturn A (previously, this was the
     case only for EEG channels).
2. We fixed a bug in the *p*-value calculation for Table 2 in the main paper.
   However, since we did not make any claims about the p-values for any
   specific channels, our interpretation of the results remains relatively
   unchanged.
3. We added details on (group session, task) combinations that we did not use
   in the computation of the results in Table 2, along with the reasons for
   excluding that data from the analysis.

---

### Author Response · Authors · 2023-08-31
**Update (2023-08-30)**

# Update (2023-08-30)

We have made the following updates to our manuscript.

## Update to valence/arousal prediction results.

We have updated Table 1 with results from updated experiments on valence and
arousal prediction from brain data. Notably, our accuracy for predicting
valence and arousal has **increased dramatically, from $\approx$ 63% to
$\approx$ 88%.**. We made the following changes to achieve this:
- *LSTM configuration*: In our original model, we mistakenly configured
  the LSTM with a single output node, which was intended to predict both arousal
  and valence. This led to the reduced classification accuracy reported in the
  previous version of the paper. We have now reconfigured the LSTM to have two
  distinct output nodes: one for arousal and another for valence. With this
  architecture, we observe a significant improvement in
  classification accuracy for the fNIRS data.
- *Batch size*: We increased the batch size from 512 to 1024.

We have also accordingly updated the reproducibility checklist in the
supplementary material, as well as Figures 2 and 3 to reflect the above
changes. We also updated the reproducibility checklist in the supplementary material
with the actual number of EEG and fNIRS
channels used (this information was already in the main paper).

Finally, the last two columns of Table 1 in the last version of the manuscript
had identical values. However, this was not quite representative of what our
model actually does---it computes the sum of the losses from both valence and
arousal. Table 1 in the updated manuscript now reflects this.

### Comparison to SOTA

While the goal of our experiment was not to beat SOTA benchmarks, it is
nevertheless heartening to see that our mean prediction accuracy is higher than
that of any of the works we surveyed save for the one by Trambaiolli et al.
(2018), which is comparable to ours in terms of performance once error margins
are taken into account.

## Addition of cross-validation error information to Table 2.

We have added columns to Table 2 reporting our mean error and standard error of
the mean for the leave-one-out cross-validation procedure used to generate the
results in this table.

## Minor rewording in Introduction

In the light of reviewer W3oi's [latest
feedback](https://openreview.net/forum?id=ZJWQfgXQb6&noteId=NKjm2WhrKi), we
have updated lines 30--32 of the manuscript to emphasize that the focus of our
work is not specifically on search-and-rescue missions, but rather on the
interesting social behaviors that can be observed in complex collaborative
tasks.

## References

L.R. Trambaiolli, C.E. Biazoli, A.M. Cravo, et al. Predicting affective
valence using cortical hemodynamic signals. In: Scientific Reports 8 (2018),
p. 5406.

---

### Decision · Program_Chairs · 2023-09-22

**Decision:**

Accept (Poster)

**Comment:**

This paper describes a novel dataset that aims to enable researchers to unpack the complex relationship between internal states and externally observable behaviour in settings involving team dynamics.

The work is novel and impressive, especially given the amount of modalities that are measured.

Quite a few of the reviewers self reported quite low confidence on the rating of the paper so I took a closer look myself.

I would recommend to accept the paper. However, my biggest fear right now is that the presentation of the work may make it hard for the ML community to really engage with the material in the way that this dataset deserves. I think this would be greatly helped with more hints as to how the dataset could be used.

There are responses to reviewers (not listed exhaustively) e.g.
"In more detail, we add that the experiments presented in this paper are exploratory (i.e., meant to recognize novel patterns in the data, which can potentially lead to the generation of new hypotheses) rather than confirmatory (i.e., testing existing hypotheses). Both types of experiments are required for progress in science [1]. Exploratory experiments are especially appropriate when the topic of research---in our case, machine learning based on brain data---is relatively less well-studied compared to other modalities (due to the inherently challenging nature of collecting brain data) such as images, text, and audio."
On l 83-85 of the paper, this is alluded to but not expressed clearly enough for the ML community.

Some more explicit references for the ML community reflected in the introduction of the paper could really help in improving the impact of the work. Where perhaps a little too much emphasis is placed broad social science aspects without a strong link back in the experiments themselves - I think more needs to be done in explaining how the ToM aspects can be investigated - is it purely about affect and the neural correlates with respect to expressed behaviour?  How would someone investigate intentions? With what data? I know you are tight on space and sometimes you don't want to give too many ideas if you are worried about being scooped but please consider providing some breadth of suggestions in the appendix (if no space in the main paper can be found).

Regarding the cross-validation, it is not clear if these were carried out by leave one subject/group out or by sample. In principle, having a baseline that assumed IID data even across samples of the same subject is ok as long as this is acknowledged. A perhaps more reliable approach would be do person-independent training. However, that may not be apropriate for brain data; I don't know.

Different strategies may be used in any case depending on what your research question is. However, not stating it is extremely problematic. Engineers who may take this work and run their own experiments for actual applications may be severely overfitting without realising. Being mindful of this when communicating the results is crucial.